# Investigating Moral Evolution Via LLM-Based Agent Simulation

## Abstract

The evolution of morality presents a puzzle: natural selection should favor self-interest, yet humans developed moral systems promoting cooperation. We introduce an LLM-based agent simulation framework modeling prehistoric hunter-gatherer societies with agents of varying moral dispositions based on expanding circles of concern. The framework demonstrates how moral dispositions interact with environmental pressures and cognitive constraints to produce different evolutionary outcomes. Our approach offers four key contributions: methodologically, it enables psychologically realistic evolutionary simulations; theoretically, it reveals the critical role of cognitive factors in moral evolution; empirically, it provides evidence for how different moral orientations succeed under varying conditions; and programmatically, it establishes an extensible simulation framework for investigating diverse social evolutionary questions. This work establishes a novel complementary paradigm to traditional evolutionary biology and anthropological research for investigating complex social evolution.

## 1   Introduction

The emergence and evolution of morality represents one of the most enduring puzzles in evolutionary biology and social sciences (Haidt, 2007; Greene, 2013). From an evolutionary standpoint, natural selection should favor individuals who maximize their reproductive success, often through selfish behaviors that increase resource acquisition at others' expense (Dawkins, 1976). Yet, humans and some other species have evolved complex moral systems that frequently promote cooperation, altruism, and other prosocial behaviors that can seemingly contradict individual fitness maximization (Tomasello, 2016). This apparent contradiction presents a profound scientific question: Under what conditions does morality provide an evolutionary advantage?

Prior research has approached this question through multiple complementary paradigms. Evolutionary game theory has provided mathematical frameworks demonstrating how strategies involving reciprocity, punishment, etc., can yield higher payoffs than pure selfishness under specific conditions (Nowak, 2006; Axelrod & Hamilton, 1981). Anthropological research has documented cross-cultural moral universals alongside cultural variations, suggesting a complex interplay between evolved predispositions and cultural learning (Henrich, 2015; Curry et al., 2019). Evolutionary biologists have proposed various mechanisms such as kin selection (Hamilton, 1964), reciprocal altruism (Trivers, 1971), and group selection (Wilson & Wilson, 2007) to explain how seemingly altruistic traits might evolve. Some moral frameworks have identified key patterns in moral cognition and behavior, such as the expanding circle of moral concern from self to kin to larger social groups (Singer, 1981), and the fundamental moral dimensions including care, fairness, loyalty, authority, and sanctity (Haidt, 2007).

While these approaches have yielded valuable insights, they face significant methodological limitations when investigating the full cognitive and behavioral dynamics of morality evolution. Traditional mathematical models necessarily abstract away the rich complexity of human cognition and interaction with the environment. Evolutionary biologists and anthropological studies provide observations and hypothesis but cannot directly test causally.

Descriptive moral frameworks illuminate current moral patterns but cannot directly observe their evolutionary development.

Recent advances in Large Language Models (LLMs) present a novel methodological opportunity to address these limitations (Park et al., 2023; Aher et al., 2023). LLM-based agent simulations can model entities with sophisticated cognitive architectures—including values, memory, perception, reasoning, and social dynamics—that generate emergent, complex behaviors (Park et al., 2023; Horton, 2023). This simulation paradigm allows us to observe interactions between moral cognition, behavior, and evolutionary outcomes under controlled conditions while providing rich, realistic psychological details that surpass traditional agent-based models (Aher et al., 2023; Liu et al., 2023).

In this paper, we introduce a novel LLM-based agentic simulation framework for investigating the evolution of morality in a simulated prehistoric hunter-gatherer environment. Our simulation framework successfully models agents with distinct moral dispositions through a sophisticated cognitive architecture. As evidenced by our validation experiments, these agents reliably exhibit behaviors faithful to their assigned moral types. Furthermore, we conduct a series of experiments to simulate moral evolution under different settings, including resource scarcity, moral type unobservability, high communication cost, etc. These experiments show that morality can generally promote cooperation and therefore improve evolutionary advantage, but the success of cooperation is also greatly affected by different setting parameters, such as resources, communication cost, identifiability, etc., providing insights for researchers to further investigate the complexity of this issue.

Our research makes four distinct contributions to the study of morality and human evolution. First, methodologically, we develop a novel computational approach using LLM-based agents that enables psychological realism in evolutionary simulations—a capability unavailable in previous mathematical or game-theoretic frameworks. Second, theoretically, we advance the understanding of morality's evolutionary dynamics by demonstrating how moral dispositions interact with environmental pressures, cognitive limitations, and social structures to produce stable or unstable evolutionary outcomes. Third, empirically, we provide specific evidence for the evolutionary advantages of different moral dispositions under varying environmental conditions, cognitive constraints, and social configurations. Fourth, programmatically, we establish an extensible agentic simulation framework MORE and environment simulation platform SOCIAL-EVOL that enables further investigation into diverse social evolutionary questions, from norm emergence to reputation systems to inter-group dynamics.

## 2 Related Work

### 2.1 Evolutionary Origins of Morality

Evolutionary biologists have proposed various mechanisms to explain how seemingly altruistic traits might evolve. Theories of kin selection (Hamilton, 1964) and reciprocal altruism (Trivers, 1971) show how limited forms of cooperation could evolve among relatives and repeated interaction partners. More recently, cultural group selection theories (Boyd et al., 2011; Henrich, 2015) explain how groups with stronger moral norms outcompeted others, leading to the genetic evolution of psychological predispositions supporting moral behavior.

Evolutionary game theory has provided mathematical frameworks demonstrating how cooperation can evolve under some strategies similar like previously mentioned mechanisms by showing strategies can yield higher payoffs than pure selfishness under specific conditions. Nowak's "five rules for the evolution of cooperation" identifies key mechanisms: kin selection, direct reciprocity, indirect reciprocity, network reciprocity, and group selection (Nowak, 2006).

While these works provide great insights and a mathematical foundation for cooperation and why morality could possibly evolve, it abstracts away the rich complexity of human cognition for us to get a full picture of the dynamics of moral evolution.

## 2.2  Moral Frameworks

Our agent design draws upon descriptive moral frameworks that characterize the psychological and behavioral essence of morality. Moral Foundations Theory (Haidt, 2007) identifies five fundamental moral dimensions: care/harm, fairness/cheating, loyalty/betrayal, authority/subversion, and sanctity/degradation. The Theory of Dyadic Morality (Gray et al., 2012) emphasizes harm as the root of morality, while Morality-as-Cooperation theory (Curry et al., 2019) identifies seven cooperative behaviors as essential: helping kin, helping group members, reciprocating, being brave, deferring to superiors, dividing resources fairly, and respecting others' property.

The Expanding Circle Model (Singer, 1981) conceptualizes morality as a process of empathetic concern expanding from self to kin to larger social groups. We find that the concept of 'group circle' in this model provides an elegant framework for distinguishing more moral from less moral agents, as the applicability scope of moral characteristics from previous theories maps naturally onto the scope of group concern. An agent who cares only about themselves cannot meaningfully engage with most moral characteristics—care for others, group welfare, fairness, reciprocity, loyalty, and respect for authority remain inapplicable. When an agent extends their concern to kin, a subset of moral characteristics becomes relevant, including care for others, loyalty, and respect for authority within the family unit. When an agent's concern encompasses the broader group beyond kinship, the full spectrum of moral characteristics becomes applicable, enabling the most comprehensive expression of moral behavior. If one treats as his group only others who also treat him as their group, this naturally incorporates reciprocity and fairness principles. In this way, the key evolutionary mechanisms identified by biologists—kin selection, group selection, and reciprocal altruism—are naturally incorporated into the expanding group circle model.

This natural mapping between moral characteristics and group circles offers a powerful theoretical foundation for our simulation, enabling systematic investigation of how different levels of moral concern affect evolutionary outcomes while maintaining theoretical consistency with established moral frameworks.

## 2.3  LLM-Based Agent Simulation

Recent advances in large language models (LLMs) provide the methodological foundation for our approach. LLM-based agent simulations can model entities with sophisticated cognitive architectures—including values, memory, perception, reasoning, and social dynamics—that generate emergent, complex behaviors (Park et al., 2023; Horton, 2023).

Prior work demonstrates this approach's versatility across domains: Park et al. (Park et al., 2023) created interactive simulations with complex social behaviors, Horton (Horton, 2023) applied LLM agents to economic simulations, and Aher et al. (Aher et al., 2023) validated that LLM agent simulations can reproduce human behavioral experiment results.

Wang et al.'s "Artificial Leviathan" (Wang et al., 2023b) explores social order in LLM-based agent societies, assuming all agents are inherently selfish and focusing on how social order emerges from this assumption. Our work takes a fundamentally different approach by explicitly modeling agents with varying moral dispositions—a design choice that better reflects the diversity of human moral psychology.

LLM agents offer several key advantages for studying morality evolution: they generate emergent social dynamics from individual-level cognitive processes (Liu et al., 2023); they allow controlled experimentation with variables impossible to manipulate in real-world settings; they provide transparent access to agents' decision-making processes; and they can simulate long timescales of social development (Bansal et al., 2023). Our work represents the first systematic application of LLM-based simulations to investigate the evolution of morality in prehistoric human societies, where moral systems likely first emerged.

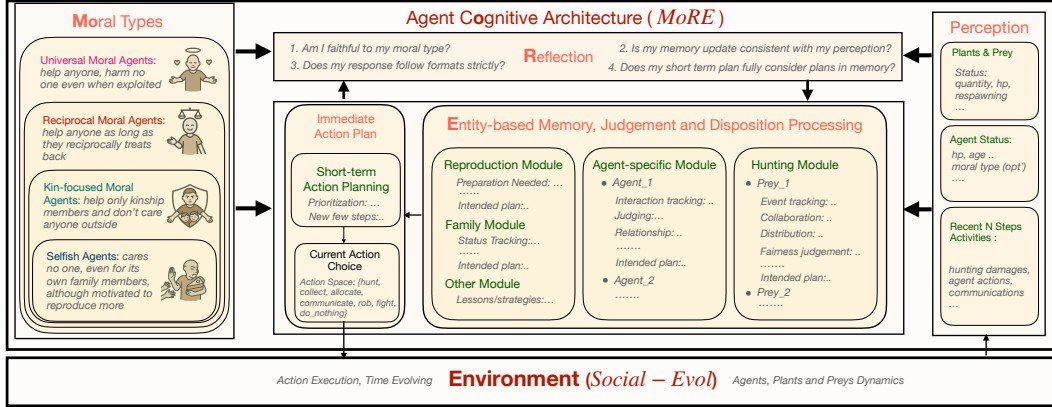

Figure 1: Overview of our simulation framework. The MORE agent architecture comprises three primary components: (1) a moral value module prescribing one of four moral types based on expanding circles of concern; (2) a perception module processing environmental information; and (3) entity-oriented cognitive modules that update memory, form judgments, and generate action plans consistent with the agent's moral type around entities. Before execution, agents perform self-reflection to verify consistency with observed facts and moral dispositions. The action execution will be send to SOCIAL-EVOL environment that updates the environment status automatically and trigger the next round of agent perception-action cycle.

## 3 Framework

### 3.1 Simulation Environment

Our simulation environment SOCIAL-EVOL creates a text-based prehistoric hunter-gatherer society in which agents navigate resource constraints, environmental challenges, and social dynamics. This environment enables investigation of the evolutionary pressures that likely shaped early human moral systems.

**Survival Mechanics** Agents maintain health points (HP) that diminish over time and from injuries, requiring replenishment through resource collection. Agents face a maximum lifespan constraint modeling natural senescence.

**Production and Reproduction Mechanics** The environment contains plants (low-risk, low-reward) and animals (high-risk, high-reward). Hunting success depends on physical power differentials, with failed attempts causing injury. This creates natural collaboration incentives. Agents meeting age and HP thresholds can reproduce, with offspring requiring parental investment for survival.

**Social Interaction Mechanics** Agents can allocate HP to others, communicate for coordination, and engage in robbery or fighting. Success in aggressive actions depends on relative physical power. Unlike hunting, antisocial behavior has no automatic punitive mechanisms—victims must independently respond. This design allows agents to act according to their moral types without artificial constraints.

This environmental design creates a complex adaptive system where survival pressures, resource competition, cooperation opportunities, and communication capabilities interact to influence the differential success of varied moral dispositions. Detailed rule explanations and configuration settings appear in the supplement.

### 3.2 Agent Design

We present our agent design framework MORE that has a **m**orality driven **e**ntity-oriented cognitive processing architecture with **r**eflection capability.

#### 3.2.1 *Moral Types*

To model evolutionary pressures, we endow all agents with a shared foundational value: maximizing survival and reproduction. Beyond this baseline, we implement varying levels of moral disposition informed by theoretical frameworks discussed in our related work.

We operationalize morality based on the "expanding circle" (Singer, 1981) concept, categorizing agents along a spectrum of moral concern:

*Self-focused agents* care only about themselves. A definitional challenge emerges with purely selfish agents: if they care exclusively about themselves, why would they invest in reproduction? Yet defining selfish agents as those who care about offspring would conflate them with kin-focused agents. We resolve this by defining them as agents who aim to reproduce but provide no further aid to offspring. This reproductive strategy is common in nature, exemplified by r-selected species (Pianka, 1970; Stearns, 1992) such as many fish, amphibians, and invertebrates that produce thousands of eggs but provide no parental care (Gross, 2005; Reznick et al., 2002; Trumbo, 2012). These organisms maximize their reproductive success through quantity rather than parental investment in each offspring's quality.

*Kin-focused agents* extend moral concern to genetic relatives, providing care and resources to family members while treating non-kin instrumentally.

*Group-focused agents* extend moral concern beyond kin to include non-related group members. For this category, we address a key definitional challenge: who constitutes the "group" worthy of moral consideration? This leads us to distinguish between two variants. *Reciprocal group moral agents* extend care only to those who reciprocate similar moral concern, creating a self-consistent moral circle based on mutual recognition. *Universal group moral agents* extend care to all individuals regardless of their moral orientation. While superficially more expansive, this variant presents theoretical inconsistencies—violating fairness and reciprocity principles while benefiting agents who may undermine group welfare. Such agents risk exploitation by selfish individuals. We include this type because its non-violent orientation aligns with some intuitive conceptions of morality.

This framework yields four distinct moral types that enable systematic investigation of how different moral dispositions affect evolutionary outcomes. We acknowledge that this discrete categorization simplifies the continuous nature of moral concern in humans for experimental tractability.

#### 3.2.2 *Agent Cognition Framework*

Our simulation employs LLM-powered agents with a cognitive architecture comprising three primary components:

*Agent Initialization* Each agent receives a moral type profile, environmental rules, and a knowledge handbook ensuring comparable baseline understanding without privileged strategic information.

*Perception and Cognitive Processing:* The perception module processes current environmental status and recent activities. The cognitive system uses an entity-based approach that maintains memory, makes judgments, and forms dispositions around entities (other agents, prey) rather than event-based processing. This method effectively prompts LLMs to consider relevant context and perform appropriate reasoning.

*Cognitive Processing System:* We designed an integrated entiy-based system that maintains memory, makes judgments, and forms dispositions around entities like other people and hunting animals. This is in contrary to the event based cognitive processing that records a log-book like memory and decision history. Our preliminary studies demonstrate that this method effectively prompts LLMs to consider relevant context and perform appropriate

218 reasoning compared to simpler approaches. The entity-based structure provides a template
219 for identifying important information and creating narrative-like understanding.

220 *Action Planning:* This module prioritizes updated memories and dispositional plans to
221 formulate specific actions. This is crucial because the simulation environment may contain
222 many entities toward which an agent might have multiple intended interactions.

223 *Reflection Module:* This verification component ensures cognitive processing and action
224 planning remain consistent with factual information and faithful to the agent's moral type,
225 while producing properly formatted responses.

### 3.3 Operation Cycle

227 The simulation operates as a sequential process where agents and the environment interact
228 in defined steps: *Environment Update*, where the simulation refreshes resource availability,
229 agent status changes, and advances time; *Agent Perception*, where each agent receives
230 observations about current environmental state and recent activities; *Cognitive Processing*,
231 where agents use their architecture to process perceptions, update memory, form judgments,
232 and develop dispositional plans consistent with their moral type toward different entities
233 (prey or other agents) or goals (reproduction etc); *Action Planning*, where agents need to
234 consider their dispositional plans and conditions to prioritize and make specific action
235 plans for next few steps; and *Consequence Resolution*, where outcomes of all actions are
236 determined. This cycle repeats continually, enabling emergent complex social behaviors
237 while maintaining tractable simulation parameters. The LLM serves as the cognitive engine
238 for each agent, providing reasoning capabilities necessary to navigate moral dilemmas, form
239 social strategies, and respond to environmental pressures in ways that reflect human-like
240 cognitive processes.

### 3.4 Simulation Analysis Assistant Agent

242 Throughout our project development, we identified a significant challenge in LLM-based
243 agent simulations: interpreting the vast quantities of generated data. While having rich,
244 multidimensional data offers tremendous analytical potential, extracting meaningful in-
245 sights from this complexity requires specialized methodological approaches. To address
246 this challenge, we developed a simulation analysis assistant agent that serves two critical
247 functions. First, it automatically generates comprehensive statistical reports containing
248 the key metrics visualized in our figures. Second, we implemented a series of function
249 calls to enable an interactive Q&A ability when user uses a readily available code copilot
250 agent like Copilot or Cursor. It can allow researchers to interrogate specific agent behaviors,
251 motivations, and decision processes (e.g., "Why did Agent X perform action Y?"). This
252 analytical tool has proven invaluable for understanding simulation dynamics and iteratively
253 refining our agent design architecture. We provide detailed specifications of this system in
254 the supplement.

## 4 Experiments

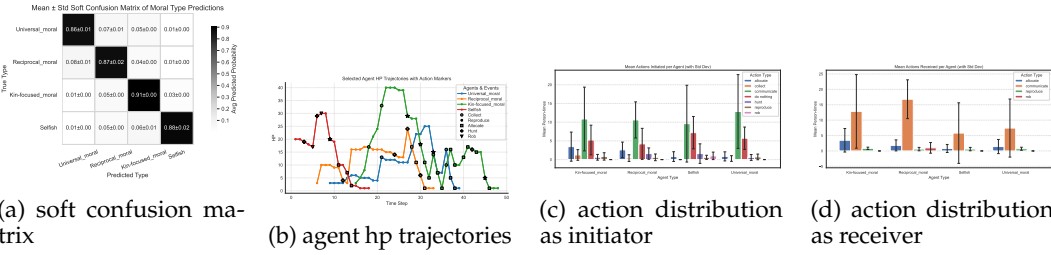

(a) soft confusion matrix

(b) agent hp trajectories

(c) action distribution as initiator

(d) action distribution as receiver

Figure 2: Major baseline experiment.

256 The agent simulation runs are using OpenAI's GPT-4.1-mini API.

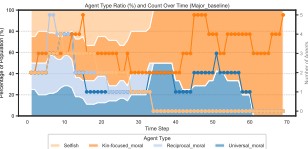

(a) Population and type ratio for major baseline test

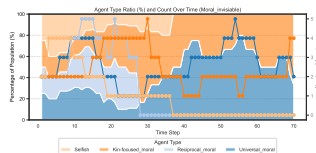

(b) Population and type ratio for invisible morality test

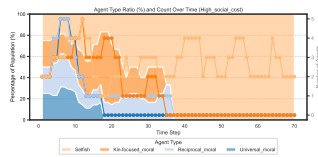

(c) Population and type ratio for high social cost

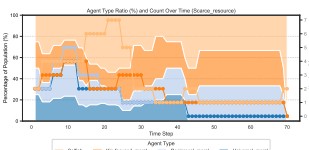

(d) Population and type ratio for scarce resource

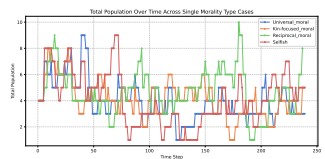

(e) Population for single agent case

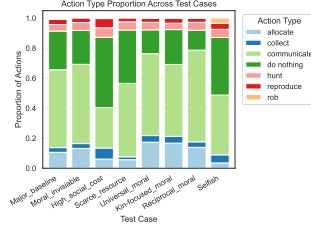

(f) Action Type Proportion Across Experiment Settings

Figure 3: Comparison across 8 experimental configurations.

## 4.1 Validation Experiments

Before we ablate the factors that affect the evolution of morality, we first validate the design of our LLM-based agents to check if it can simulate realistic behaviors faithful to its moral type.

To test it, the major criteria is to ensure that one can infer an agent's moral type based on the observed behaviors and the cognitive processing. To obtain a quantitative results, we run a baseline experiment and ask advanced models like GPT-4.1 to perform a moral type inference task and output its classification distribution for 3 times. By averaging the judgment results over every moral type in a simulation, we can obtain a confusion matrix of the advanced model along with the standard deviation. As shown in Figure 2(a), the advanced models can achieve high accuracy in inferring the moral type of agents reliably with the standard deviation less than 0.02, showing a reliability of our simulation.

Further, we show the distribution difference of the actions among different moral types. As shown in Figure 2(c) and (d), where the former shows the distribution of actions performed by the agents of each type, and the latter shows the distribution of received actions by the agents of each type, the actions are distributed differently among different moral types, with violent actions like robbing and fighting disproportionally occurs more in more selfish agents, and communication and allocation signficantly more in more moral agents, which indicates that our agents can simulate realistic behaviors.

To give a sense of one agent's life events, we randomly pick one agent from each moral type in the baseline experiment and show their HP curve along with the actions they perform in Figure 2(b). The kin-focused moral agent demonstrates a life marked by strong family bonds and self-sacrificial behavior. Born at step 15 with 3 HP, it gained HP through two lucky big collections, then delivered two children followed by a series of sharing behaviors. The agent's HP curve shows regular fluctuations as it balanced between self-preservation and family support. After a third reproduction, however, it did not manage to gain enough energy in time and died. Its life was characterized by frequent communication with family members, seeking alliances for mutual survival, and prioritizing children's well-being over personal gain.

We also show the evolution dynamics in environments with agents of the same moral type. As shown in Figure 3(e,f), an obvious pattern is that the selfish agents frequently drops its population down to only one person, showing a sense of difficulty to co-exist. While the moral agents rarely see such near-extinction phenomena.

## 4.2 Main Experiments

### 4.2.1 Experiment Factors and Settings

We systematically investigate the influence of both social and environmental factors on moral evolution. By controlling both agent capabilities and environmental conditions, we isolate key variables affecting evolutionary outcomes. Our experimental design varies four critical dimensions:

**Baseline Setting** Our baseline configuration employs a non-scarce resource, low social interaction cost, and direct moral type observability to provide a relatively easy mode of survival. The progression of the population ratio of each moral type is shown in Figure 3(a). As we can see, the kin-focused moral agents ends up dominating the population. This is because kin-focused agents dont' have much collaboration distribution issue, and when the resources are not scarce, they become quickly powerful by delivering more offsprings.

**Resource Scarcity** We manipulate resource parameters (quantity, spawning rates, nutritional yield, and acquisition difficulty of plants and prey) through an integrated abundance variable that proportionally adjusts these parameters. For this experiment, we simulate a scarce setting of the resources to see how that triggers the evolution of different moral types. This experiment see a draw among kin-focused, reciprocal and selfish, with selfish eventually wins. But looking into the close dynamics, it actually shows that more moral agents are actually more competitive in the process, but the selfish agents moved fast and took a lot of resources at the beginning, getting a head lead that eventually leads to its survival. This experiments shows the complexity in the competition - many factors are involved and unnoticed ones can be fatal.

**Social Interaction Cost** To model the differential temporal scales of social versus productive activities, we implement adjustable time costs for social interactions. Our framework allows varying numbers of social interaction rounds (communication, fighting) before agents can undertake resource acquisition or reproduction. This mechanism enables flexible control over the relative investment required for social engagement versus production. This experiment investigates in a high social interaction cost by allowing only 1 social interaction round before production, making the communication extremely hard. As we can see in Figure 3(c), the selfish agents clearly dominates the population. The reason is that the high communication cost makes the collabortion extremely hard, so moral agents spend more time to get together to take productive actions. Meanwhile selfish agents just go take actions directly, obtaining an advantage.

**Moral Type Observability** The ability to accurately identify others' moral dispositions represents a critical cognitive capability affecting cooperation dynamics. When moral types are directly observable, agents can avoid misattributing intentions and form more stable cooperative relationships. We investigate how this observability capability influences which moral types achieve evolutionary dominance under otherwise identical conditions. As shown in Figure 3(b), the kin and universal moral agents stays in the end while others die out early. Reciprocal moral agents did not survive because he was mistaken as a selfish agent due to misjudgement. This shows that the ability let others know your true moral type is critical for the survival of moral agents.

## 5 Discussion

**Insights into Prehistoric Societies and Evolutionary Theories** Our simulations reveal that kinship-focused agents often dominate when moral type perception is limited, providing insight into the prevalence of matrilineal systems in prehistoric societies (Holden & Mace, 2003; Mattison et al., 2011; Wang et al., 2023a). Without genetic verification methods, maternal relationships offered the only unambiguous biological connections, creating a reliable foundation for cooperative groups (Hamilton, 1964). This extends to ethnic identity formation—our findings suggest that successful family-based cooperative groups could eventually dominate population genetics (Soltis et al., 1995), with cultural mechanisms

emerging to maintain cooperation as groups expanded beyond immediate recognition thresholds (Boyd & Richerson, 1987; McElreath et al., 2003).

**Insights for Moral Theory** Our findings support the expanding circle model as a unifying moral framework. Empirical results confirm that broader, yet self-consistent moral circles generally produce superior evolutionary outcomes. The model elegantly integrates diverse moral characteristics while naturally incorporating evolutionary strategies. Additionally, our work highlights how cognitive factors—particularly the reliability of in-group identification mechanisms—critically affect moral evolution, potentially explaining the emergence of specific moral norms that facilitate reliable group recognition.

**Connection with More Theories** Our simulation yields more phenonmena that can be connected to a wide range of theories. Communication imposes coordination costs, forcing agents to balance social interaction and resource acquisition, consistent with bounded rationality theory (Simon, 1991). Misunderstandings, stemming from limited behavioral observation, often lead to conflict, echoing communication theory on information transmission limits (Shannon, 1948; Deutsch, 1973). Universal moral agents are exploited when they never retaliate, underscoring the role of altruistic punishment in sustaining cooperation (Fehr & Gächter, 2002). Moral agents also face dilemmas where acting alone may be more beneficial than collaborating, illustrating the trade-offs between cooperation and individual fitness (Bowles & Gintis, 2004). These dynamics emerge naturally in our simulations, offering a unified framework for social evolution theories typically studied in isolation. Further theoretical connections are detailed in the supplement 1.

## 5.1 Limitations and Future Work

Our study presents several limitations that suggest directions for future research:

First, we emphasize that our simulation approach is not claiming to definitively answer why morality evolves. We present this method as a *complementary* tool to traditional anthropological and evolutionary biology research, providing rich detail and enabling study of factor interactions. Understanding these limitations is crucial for proper application.

Second, our categorical operationalization of moral types (self, kin, group) simplifies the continuous nature of moral concern in real human cognition. Humans typically distribute varying degrees of moral weight across concentric circles rather than exhibiting categorical boundaries. Future work should implement continuous moral weighting distributions.

Third, by abstracting away spatial and temporal constraints, our simulation sacrifices ecological validity for computational tractability. Spatial proximity fundamentally shapes interaction patterns in human societies, and implementing meaningful spatial constraints would likely yield additional insights into moral evolution dynamics.

Fourth, our framework omits mate selection mechanisms—a central feature of biological evolution with substantial implications for moral behavior. Incorporating partner choice dynamics would likely enhance prosocial behavior toward non-kin as agents seek to demonstrate desirable moral traits to potential mates.

## 6 Conclusions

We have presented an LLM-based agent simulation framework for investigating moral evolution in prehistoric hunter-gatherer environments. Our experiments demonstrate that different moral dispositions achieve varying evolutionary success depending on environmental and cognitive factors. Key findings include the dominance of kinship-focused morality when moral type perception is limited, the advantage of selfish strategies under high communication costs, and the importance of reliable group identification mechanisms for broader moral circles to evolve. Our results support the expanding circle model as a unifying framework for understanding moral evolution while providing insights into prehistoric social structures. This approach establishes a novel paradigm for investigating social evolutionary dynamics that can be extended beyond morality to other complex social phenomena.

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

# Appendix

# Contents

## A  General Discussions

### A.1  Code Release

Our code is released under the MIT license. The code is available at `https://anonymous.4open.science/r/Social-Simulation-with-Moral-Agents-94B5`. This platform will be actively maintained and updated to support more features to support more research questions. We welcome any collabortion, contribution, feedback and feature requests.

### A.2  Ethical Consideration

Our project is, at its core, a simulation study of ethics itself. As such, it does not raise the typical ethical concerns associated with methodological research that might be misused. Importantly, our findings can be interpreted as supporting the general proposition that morality is beneficial for humans. The factors that sometimes cause moral agents to fail in evolutionary competition can, in fact, offer valuable insights for promoting social causes and designing mechanisms to enhance the evolutionary advantage of moral individuals. However, we caution against the simplistic interpretation that the conditions under which moral agents fail to prevail are evidence that morality is not advantageous for humans. Such a view is an oversimplification. First, modern society differs profoundly from prehistoric hunter-gatherer contexts. Humans have evolved to be born with moral dispositions (Hamlin et al., 2011; Bloom, 2013; Warneken & Tomasello, 2006). Also in contemporary human societies, almost no one can survive without collaboration, promoting moral behaviors. Second, it is crucial to understand the specific causal role that morality plays in success or failure. For example, our results show that when communication is prohibitively costly, moral agents may be outcompeted by selfish ones. This occurs because morality often inclines agents toward collaboration, which may not be optimal in situations where cooperation is particularly costly. However, morality does not require agents to cooperate indiscriminately; moral agents could, in principle, maintain their moral disposition while choosing to act independently when cooperation is not advantageous, and then collaborate when conditions improve. As revealed by our simulation and common wisdom, being moral does not guarantee success in every circumstance, but a lack of morality fundamentally constrains one's potential for success.

### A.3  More Discussion Over Our Methodology

As we have emphasized, our method should be viewed as a complement to traditional mathematical models, not a replacement. By incorporating rich psychological realism into the simulation, our approach enables researchers to investigate how numerous factors interact in complex ways. However, this increased realism also means that simulation results are sensitive to the specific details of these factors and may not yield the definitive answers that highly abstract mathematical models can provide.

Yet, definitive answers are not always the primary goal of research, especially in the social sciences. Often, the objective is to uncover previously unnoticed factors that influence a

phenomenon or to explore the intricate interplay among multiple variables. Such goals are difficult to achieve with traditional mathematical models, which require all relevant factors to be known or assumed in advance. Historically, researchers have relied on field studies to observe human behavior and identify these factors, but simulation now offers a cost-effective means to assist in discovery and hypothesis generation, potentially accelerating progress in the field.

Moreover, when the number of interacting factors becomes too great for analytical calculation, simulation becomes indispensable. While simulations inevitably deviate from reality—just as any modeling method, and such deviations may be amplified in large-scale runs—they can still provide valuable insights into research questions. Simulations can reveal what is possible, and the underlying mechanisms and developmental dynamics they expose may remain relevant even if the precise outcomes differ from those observed in the real world.

### A.4 Flexibility of the Simulation Platform

Our platform is designed to be flexible and extensible. By varying the configuration settings, one can use the same platform to study different research questions. For example, we have used the same platform to study the effect of different moral types, different resource distributions, different communication costs, etc. In the below section, we also list a list of findings that are connected to different research areas that could possibly be investigated further with our platform.

Moreover, we support researchers to extend beyond moral related value dispositions. One can flexibly define the value dispositions of the agents by writing appropriate prompt templates. For example, one can define agents to be of different cultural backgrounds, different religions, different political views, etc. Or one can also study the effect of specific social norms by prescribing the agents to follow certain social rules, e.g always equal distribution VS always contribution based distribution etc. Hunting-gathering environment equipped with general social interaction dynamics is very general to support a wide range of research questions.

## B    Discovered Phenomena That Connects to Other Theories

As mentioned, one key feature of what our platform can provide is that we can naturally see a lot of emergent phenomena that matter for social evolution regarding morality. These phenonmena were abstract away in the traditional mathematical models. But in our platform they will surface on their own to deepen our understanding. Those phenomena or topics were traditionaly a subject of research areas on their own, but now we can study them in a unified framework.

We list some of the observed phenomena and identified some of the theories that are related to them in Table 1. This list is definitely not exhaustive. We wish this can provide a good starting point for future researchers to discover more phenomena and theories.

We also encourage researchers to use our platform as a new way to study these phenomena and theories.

Table 1: Discovered Phenonmena and Related Theories

| Phenomena Findings from Experiments | Related Theories |
|---|---|
| Coordination is costly:
• Communication takes time and can reduce the time for other important things. | *Coordination Cost Theory* (Simon, 1991)
"Organizations face bounded rationality where coordination costs limit optimal decision-making" |
| Misunderstandings can lead to major conflicts:
• Agents may misinterpret others' intentions or actions, leading to unnecessary conflicts.
• Limited communication can cause agents to make incorrect assumptions about others' moral types or goals. | *Communication Theory* (Shannon, 1948)
"Information transmission is inherently imperfect, leading to potential misunderstandings and conflicts"
*Conflict Resolution* (Deutsch, 1973)
"Many conflicts arise from misperceptions and misunderstandings rather than actual incompatible goals" |
| Moral judgment based on actions:
• Agents evaluate others' morality by observing how they treat third parties.
• Actions toward others, not just toward oneself, shape moral reputation. | *Moral Judgment Theory* (Haidt, 2001)
"People make rapid moral judgments based on observed behaviors and their emotional responses"
*Impression Formation* (Asch, 1946)
"Observers form impressions of others' character based on their actions toward third parties" |
| Universal moral agents get exploited:
• Agents who never retaliate or punish others' bad behavior become targets of exploitation.
• Their unconditional cooperation makes them vulnerable to free-riders. | *Altruistic Punishment* (Fehr & Gächter, 2002)
"Cooperation requires punishment of defectors; pure altruism without retaliation is vulnerable to exploitation"
*Strong Reciprocity* (Bowles & Gintis, 2004)
"Evolutionary success requires both cooperation and punishment of non-cooperators" |
| Group membership is contested:
• Agents might not agree who are in the group that can share resources. | *Social Identity Theory* (Tajfel, 1979)
"Group boundaries are fluid and contested, with membership determined by shared identity markers and mutual recognition" |
| Distribution methods are complex:
• How to distribute? Distribute evenly, based on contribution, harm taken, need, can affect both the success of the end result and each other's judgement. | *Distributive Justice* (Konow, 2003)
"Fairness judgments depend on multiple principles including equality, need, and contribution" |
| Careful planning is important:
• Reproduction schedule is important. Too frequent can cause both parents and children to die. | *Life History Theory* (Kaplan et al., 1992)
"Organisms face trade-offs between current and future reproduction, with timing being crucial for survival" |
| Tendency to cooperate can sometimes have negative effect:
• Moral agents have a tendency to collaborate to acquire resources, but in some particular setting (with competition, resource being in some way), taking faster action instead of collaboration may be more crucial.
• Moral agents tend to agree to collaboration to hunt, but they might not be in a good position to hunt. | *Cooperation Dilemmas* (Bowles & Gintis, 2004)
"Cooperation can be maladaptive when individual action would yield higher returns" |
| Moral agents' mutual dependency sometimes leads to disaster end:
• Moral agents tend to trust others to help them later, but the others may also think the same and none have the extra capacity to help. | *Trust and Cooperation* (Fehr & Gächter, 2002)
"Altruistic punishment can maintain cooperation but may lead to cascading failures when trust is misplaced" |
| Mutual reinforcing / social pressure:
• When some agents reproduce, others feel compelled to do so too even their HP was not very high. | *Social Learning Theory* (Bandura, 1977)
"Social learning and imitation can lead to behavioral contagion even when not optimal for individuals" |

## C System Design Details

### C.1 Simulation Pipeline

The general system workflow functions as Fig. 4. System first initializes the environment based on the system setting config (e.g see Table 9) or resume from previous expriment run. The specific initialization phases are shown in Table 2.

Then the system enters in to an execution cycle that allows agents to perceive and perform cognitive processing to plan for actions and update the environment accordingly. The execution phases are shown in Table 3. Within this cycle, there is also a system validation and correction cycle over the agent's reponse and action to ensure its format and content are legal (see Fig. 5 and Appendix C.1).

Please refer to those tables and figures for more details.

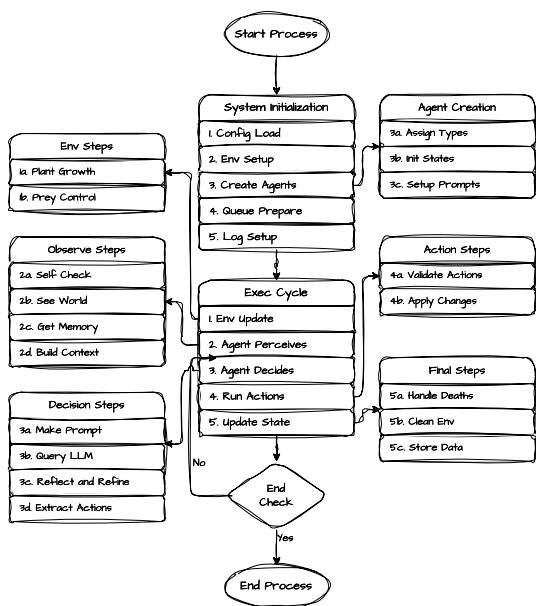

Figure 4: **Simulation Pipeline Overview** showing the main components and data flow through the system architecture. The pipeline illustrates how the Singleton-based Checkpoint, modular microservices, and key simulation processes interact to maintain consistent state and flow of information.

### C.2 System Design Principles

The Morality-AI simulation is built on two core principles: centralized state management and modular architecture. A Singleton-based Checkpoint class maintains a single, authoritative simulation state, ensuring consistency, atomic updates, and easy reproducibility. This design prevents conflicting states and simplifies debugging and resuming experiments. The system adopts a microservice-inspired structure, separating major functions—such as state persistence, agent reasoning, and LLM interfacing—into independent, easily testable modules. This modularity enhances maintainability, scalability, and flexibility, allowing components to be updated or replaced without affecting the overall system. Together, these principles provide a robust and extensible foundation for complex agent-based simulations.

## Table 2: Simulation Initialization Phases

| Phase | Description |
|---|---|
| Configuration Loading & Validation | '•' Loads parameters from configuration file (prompt paths, agent types, rules, strategies)
• Validates type correctness, constraints, and completeness
• Creates authoritative configuration object for simulation |
| Environment Setup | • Plant Resources:
- Generated based on configured abundance
- Each plant gets unique ID, initial quantity, capacity, nutrition value, respawn delay
• Prey Animals:
- Initialized with unique IDs
- HP and max health sampled from Gaussian distribution
- Assigned physical ability values
• Resources placed randomly in unoccupied grid cells |
| Initial Agent Spawning | • Instantiates agents based on population size
• Assigns moral types according to configuration ratios
• Initializes attributes: HP, age, physical ability
• No initial family ties |
| Execution Queue Setup | • Creates randomized agent sequence for fair execution
• Initializes time step counter (typically 0 or 1)
• Sets up containers for agent observations |
| Logging System Setup | • Configures comprehensive tracking system
• Creates log files for:
- Global progress summaries
- Per-step execution records
- Detailed event logs
- Error diagnostics
• Organizes logs in uniquely named directories |

## Table 3: Per-Step Execution Cycle Phases

| Phase | Description |
|---|---|
| Environment State Update | • Updates plant lifecycle: restores depleted plants after respawn delay, increases quantity for non-depleted plants
• Spawns new prey in empty locations based on probability and maximum count
• Removes dead prey from the grid |
| Agent Observation | • Self-assessment: queries HP, age, inventory, physical ability, reproductive status
• Environmental perception: detects nearby resources, prey, and other agents
• Memory retrieval: accesses past observations, messages, and action outcomes
• Context formatting: structures information for LLM prompt |
| Agent Decision Making | • Constructs system message with agent persona and rules
• Builds user message with current state and context
• LLM processes context and returns proposed action
• Validates response format and structure |
| Action Execution & Validation | • Performs response & action validation
• Applies validated actions to simulation state |
| State Finalization | • Updates agent HP, inventories, and environmental quantities
• Handles communication and memory updates
• Performs system-wide consistency checks
• Records detailed logs of agent states, environment state, and metrics
• Prepares state for next cycle |
| Termination Check | • Evaluates termination criteria (max steps, population collapse, goals)
• Either concludes simulation or increments time step |

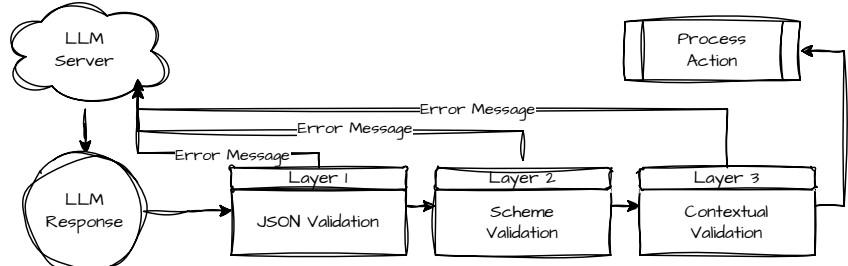

Figure 5: Multi-layer validation and retry framework showing the escalating levels of validation applied to agent actions. The diagram illustrates the three validation layers: syntactic and schema validation, contextual rule-based pre-validation, and action handler final validation, along with their respective feedback loops and retry mechanisms.

Table 4: The Checklist of Multi-Layer Response & Action Validation.

| Layer | Description |
|---|---|
| Layer 1: Syntactic and Schema | • Applied immediately after LLM output generation
• Two critical checks:
- Syntactic validation: Ensures proper JSON formatting
- Schema validation: Verifies required fields, types, and enumerations
• Retry mechanism:
- Resends prompt with error metadata
- Limited to predefined maximum attempts
• Focus: Structural correctness only |
| Layer 2: Contextual and Rule-Based | • Domain-specific validation within Agent Decision Making phase
• Contextual checks:
- Target existence and accessibility
- Location-based constraints
- HP sufficiency for action costs
• Memory constraints:
- Long-term memory capacity limits
• Feedback loop:
- Human-readable error messages
- Updated prompts with feedback
- Configurable retry rounds |
| Layer 3: Action Handler Final | • Executed during Action Execution phase
• Domain-specific validation in action handlers
• Dynamic condition checks:
- Agent adjacency for physical interactions
- HP sufficiency with current state
- Race conditions with shared resources
• No LLM retry mechanism
• Failure handling:
- Action nullification or failure processing
- Logging to agent observation history |

# D Agent Design Details

## D.1 Agent Designs and Workflows

Agents are the primary decision-making entities in the simulation. They possess a set of core attributes that govern their physical capabilities, cognitive constraints, and eligibility for specific actions (see the agent attributes in Table 9).

At the begining of agent initialization, agent will be given thier value/moral type prompt and all the system prompts like environment dynamics, requirement, commonsense strategies etc (prompt details see Appendix E.2). Then during each execution cycle, the agent will be given the perception of the environment and its own status, and perform cognitive processing to make action plan. They will perform one round of reflection before finalize their response that contains their cognitive processing and action plan. The process follows Fig. 6 to make decisions.

For the current project, the structure of agent's moral type is listed in Table 5, with the rationale of the design choices in the main text. We want to note that these moral types is not the only way to define the value of an agent. The value can be defined in many other ways - one can focus on the action principles, or calculation of utility, or even involve in culture and religion to study diffenrent problems.

The structure of agent's perception space is listed in Table 7. The structure of agent's cognition is listed in Table 6. The content in action space is listed in Table 8.

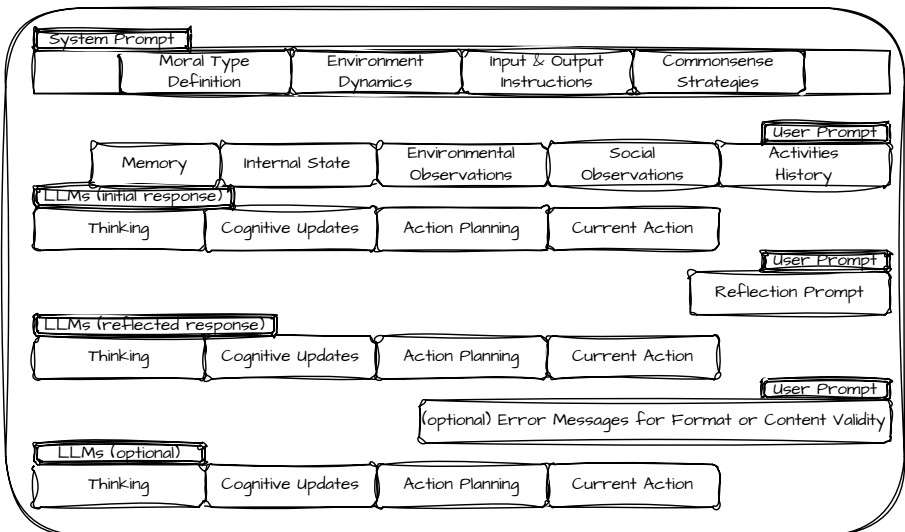

Figure 6: The LLM query process for decision making, illustrating the flow from observation gathering through prompt construction, LLM interaction, and action validation. This process shows how environmental perceptions, agent state, and memory are integrated to produce contextually relevant decisions within the simulation environment.

## D.2 The Quantitive Model for Calculating Action Results

The success rate and damage point in actions like hunting, robbery etc is calculated based on the physical ability of the agents. The physical ability values is initialized as a random number from a Gaussian distribution with specified mean and standard deviation in the configuration file Table 9 (with 0 standard deviation, there will be no random variation). Note that preys also has a physical ability value that are initialized in the same way.

| Moral Type | Core Characteristics | Expected Typical Behaviors | Expected Cooperation Pattern |
|---|---|---|---|
| Universal Group-Focused Moral | Aim for universal well-being and collective good, harm-action averse | Share resources freely; protect others from harm; communicate transparently | Highly altruistic and cooperative with all agents |
| Reciprocal Group-Focused Moral | Fairness and mutual benefit within in-group, harm action allowed | Form strong bonds with cooperative peers | Cooperative with in-group; neutral or adversarial to out-group or selfish agents |
| Kin-Focused Moral | Prioritize genetic relatives above all else, harm action allowed | Form close-knit kinship clusters; sacrifice for kin | Intensely altruistic toward family; indifferent or competitive toward non-kin |
| Reproductive Selfish | Personal reproductive success, harm action allowed | Acquire resources for own survival; opportunistic tactics | Cooperate only when serving reproductive interests; inclined to hoard resources |

Table 5: Agent Moral Types Summary. Here we summarize the core characteristics, expected typical behaviors, and expected cooperation patterns of these moral types. However, the simulated behavior for each agent might not strictly follow the expected behaviors due to the randomness of LLM's output. The exact prompt for each moral type is shown in E.2

Table 6: Agent Cognition Structure (Memory, Judgement and Planning)

| Field | Key Subfields / Description |
|---|---|
| **1. Prey Based Cognition** | • Organized by prey_id:
  ○ hunt_fact_history_of_this_prey: who hunted, effect, time step, damage, if_killed
  ○ communication_and_planning_before_killing_prey: reward, collaborators, distribution plan, objections
  ○ distribution_after_killing_prey: winner, allocation, fairness evaluation, free rider check
  ○ plan_next: next plan, retaliation plan, stage, reasoning
  ○ afterward_happenings: retaliation events, other events, lessons learned |
| **2. Agent Based Cognition** | • Organized by agent_id:
  ○ important_interaction_history: what_i_did_to_him, what_he_did_to_me (action type, success, reason, target moral type)
  ○ thinking: evaluation, judgement, relationship, agreement, plan |
| **3. Family Plan** | • Organized by agent_id:
  ○ status: how the family member is doing
  ○ plan: what to do to/with them |
| **4. Reproduction Plan** | • thinking: reasoning about reproduction plan
• preconditions_and_subgoals: specific preconditions needed
• estimated_time_to_produce_next_child: time step |
| **5. Learned Strategies** | • Lessons learned, strategies to follow in the future |

Table 7: Agent Perception Content Structure

| Category | Description |
|---|---|
| Self/Internal Information | • Current HP and health status
• Family relationships and status
• Personal attributes and capabilities |
| Environment Status | • Available plant resources
• Prey animals present in environment
• Resource locations and quantities |
| Other Agents Status | • Basic information (age, HP) of other agents
• Moral types of other agents
• Current positions and states |
| Recent History | • Last 15 steps of personal interactions:
  - Others' actions and communications toward self
  - Self's actions toward environment and others
• Recent events:
  - Changes in environment and other agents
  - Family-related news and updates
• Hunting activities:
  - Personal involvement in prey hunting
  - Related communications and outcomes |
| Memory | • Updated memory from previous step
• Immediate action plans from previous step |

Table 8: Agent Action Space Summary

| Action | Description | Requirements | HP Cost | Outcome |
|---|---|---|---|---|
| *(Re)Production Actions* | | | | |
| Collect | Gather plant resources from environment | Resource exists and is a plant node | None | Agent gains HP (quantity × nutrition value); plant quantity reduced |
| Hunt | Target prey animals for nutritional gain | Prey exists | 1 HP + additional damage if failed | If successful, prey killed and agent receives reward equal to prey's max HP |
| Reproduce | Create offspring | Minimum age and HP thresholds met | Defined in reproduction parameters | Child agent created with age 0 and initial HP, inheriting parent's archetype |
| *Social Interaction Actions* | | | | |
| Allocate | Transfer HP to other agents | Targets exist and are alive; sufficient HP | Equal to HP transferred | Recipients gain specified HP (capped at maximum) |
| Fight | Attempt to damage another agent | Target exists, is alive, not self | 1 HP resistance cost | If successful (based on ability difference), target suffers damage equal to attacker's ability |
| Rob | Forcibly transfer HP from another agent | Target exists and is alive | 1 HP + potential failure penalty | If successful, HP transferred from target to robber |
| Communicate | Send messages to other agents | Target agents exist and are alive | None | Message recorded in recipient's memory |
| *Other Actions* | | | | |
| DoNothing | Abstain from all actions | None | None | No changes to agent or environment |

**Success Rate**   The success of hunt, fight and rob actions takes on probabilistic manner. The success of such actions depends on the relative physical abilities of the involved entities. Let $\Delta PA = PA_k - PA_{target}$ represent the physical ability differential between an actor $k$ and a target entity (which could be another agent $j$ or a prey animal $A_j$). The probability of success, $P_{succ}$, for these actions is determined by the function:

$$P_{succ}(\Delta PA; I_{PA,k}, S_{PA,k}) = \min\left(\max\left((0.5 + I_{PA,k}) + 0.4 \cdot \tanh\left(\frac{\Delta PA}{S_{PA,k}}\right), 0.1\right), 0.9\right)$$

Here, $I_{PA,k}$ and $S_{PA,k}$ are agent $k$'s specific scaling parameters (an intercept offset and a slope divisor, respectively) pertinent to physical ability interactions, derived from its configuration. The function $\min(\max(x, a), b)$ ensures the probability is clipped to the interval $[a, b]$, in this case, $[0.1, 0.9]$. The outcome of such an action is then determined by a Bernoulli trial $X \sim \text{Bernoulli}(P_{succ})$.

In the descriptions that follow, $HP_k(t')$ signifies the health of agent $k$ after any initial action-specific costs have been deducted, but before other consequences of the action (e.g., gains from success, damage from failure) are applied.

**Collect**   Agent $k$ may attempt to gather resources from a designated plant node $P_i$, which possesses a current resource quantity $Q_i(t)$. The agent specifies a desired quantity $q_{req}$. For the action to be valid, $P_i$ must be a plant, and its available quantity must meet the request, i.e., $Q_i(t) \geq q_{req}$. The actual quantity gathered, $q_{coll}$, is constrained by the request, availability, and the agent's single-action collection capacity, $k_{collect}$ (a global limit):

$$q_{coll} = \min(q_{req}, Q_i(t), k_{collect})$$

A positive quantity must be collectible ($q_{coll} > 0$). Consequently, the agent's health and the plant's resources are updated as follows:

$$HP_k(t+1) = \min(\max(HP_k(t) + q_{coll} \cdot H_{plant}, 0), HP_{k,max})$$
$$Q_i(t+1) = Q_i(t) - q_{coll}$$

where $H_{plant}$ denotes the nutritional value conferred per unit of the plant resource. This action imparts no direct HP cost to agent $k$.

**Allocate**   An agent $k$ (the donor) can transfer Health Points to other agents. This is specified via an `allocation_plan`, $(h_{kj})_{j \in J}$, where $h_{kj} \in \mathbb{R}^+$ is the amount of HP designated for transfer to each target agent $j$ in a non-empty set $J \subset \mathcal{K}(t)$. The total HP intended for allocation by agent $k$ is $H_{alloc,k} = \sum_{j \in J} h_{kj}$. This action is permissible if all target agents $j \in J$ are alive and the donor possesses sufficient HP, specifically $HP_k(t) > H_{alloc,k}$. If valid, the HP of the involved agents are then adjusted:

$$\forall j \in J, \quad HP_j(t+1) = \min(\max(HP_j(t) + h_{kj}, 0), HP_{j,max})$$
$$HP_k(t+1) = \min(\max(HP_k(t) - H_{alloc,k}, 0), HP_{k,max})$$

**Fight**   Agent $k$ (attacker) may engage agent $j$ (target), provided $k \neq j$ and $j$ is alive. To initiate a fight, the attacker $k$ incurs an immediate cost $C_{fight,init} = 1$ HP:

$$HP_k(t') = HP_k(t) - C_{fight,init}$$

If $HP_k(t') \leq 0$, agent $k$ is removed from $\mathcal{K}(t+1)$. Otherwise, the outcome of the fight is determined by a Bernoulli random variable $X_{fight} \sim \text{Bernoulli}(P_{succ}(\Delta PA_{kj}; I_{PA,k}, S_{PA,k}))$, where $\Delta PA_{kj} = PA_k - PA_j$. The health point dynamics for both the target and attacker, contingent on the outcome $X_{fight}$, are:

- If $X_{fight} = 1$ (success): The target's health is reduced, $HP_j(t+1) = \min(\max(HP_j(t) - \lfloor PA_k \rfloor, 0), HP_{j,max})$.

- If $X_{fight} = 0$ (failure): The target's health remains unchanged, $HP_j(t+1) = HP_j(t)$.

In both scenarios, the attacker's health after the interaction resolves is $HP_k(t+1) = HP_k(t')$. The target agent $j$ is removed if its health $HP_j(t+1) \leq 0$.

**Rob**  Agent $k$ (robber) may attempt to forcibly extract $h_{rob,req} > 0$ HP from a target agent $j$, provided $j$ is alive and possesses sufficient health ($\text{HP}_j(t) \geq h_{rob,req}$). The robber $k$ first incurs an initiation cost $C_{rob,init} = 1$ HP:

$$\text{HP}_k(t') = \text{HP}_k(t) - C_{rob,init}$$

If $\text{HP}_k(t') \leq 0$, $k$ is removed. Otherwise, the success of the attempt is a random variable $X_{rob} \sim \text{Bernoulli}(P_{succ}(\Delta\text{PA}_{kj}; I_{\text{PA},k}, S_{\text{PA},k}))$, with $\Delta\text{PA}_{kj} = \text{PA}_k - \text{PA}_j$. Depending on the outcome $X_{rob}$, the HP updates are:

- If $X_{rob} = 1$ (success):

$$\text{HP}_j(t+1) = \min(\max(\text{HP}_j(t) - h_{rob,req}, 0), \text{HP}_{j,max})$$
$$\text{HP}_k(t+1) = \min(\max(\text{HP}_k(t') + h_{rob,req}, 0), \text{HP}_{k,max})$$

  The target $j$ is removed if $\text{HP}_j(t+1) \leq 0$.

- If $X_{rob} = 0$ (failure): No HP is transferred, thus $\text{HP}_j(t+1) = \text{HP}_j(t)$, and the robber's health remains $\text{HP}_k(t+1) = \text{HP}_k(t')$.

**Hunt**  Agent $k$ (hunter) may target a prey animal $A_j$, characterized by physical ability $\text{PA}_{A_j}$ and health $\text{HP}_{A_j}(t)$ (with maximum $\text{HP}_{A_j,max}$). The hunter $k$ incurs an initial cost $R_{hunt} = 1$ HP:

$$\text{HP}_k(t') = \text{HP}_k(t) - R_{hunt}$$

If $\text{HP}_k(t') \leq 0$, $k$ is removed. Otherwise, the outcome is governed by $X_{hunt} \sim \text{Bernoulli}(P_{succ}(\Delta\text{PA}_{kA_j}; I_{\text{PA},k}, S_{\text{PA},k}))$, where $\Delta\text{PA}_{kA_j} = \text{PA}_k - \text{PA}_{A_j}$.

- If $X_{hunt} = 1$ (success): The prey $A_j$ sustains damage $D_{A_j} = \lfloor \text{PA}_k \rfloor$, leading to $\text{HP}_{A_j}(t+1) = \max(0, \text{HP}_{A_j}(t) - D_{A_j})$. If this damage proves lethal ($\text{HP}_{A_j}(t+1) \leq 0$), prey $A_j$ is removed, and the hunter $k$ gains HP from the kill:

$$\text{HP}_k(t+1) = \min(\max(\text{HP}_k(t') + \text{HP}_{A_j,max}, 0), \text{HP}_{k,max})$$

  If the prey survives the damage, the hunter gains no HP from the hit, so $\text{HP}_k(t+1) = \text{HP}_k(t')$.

- If $X_{hunt} = 0$ (failure): The prey $A_j$ counter-attacks, inflicting $D_{prey}$ damage upon hunter $k$. This $D_{prey}$ is a characteristic of the prey (e.g., its counter-attack strength). The hunter's health is updated to

$$\text{HP}_k(t+1) = \min(\max(\text{HP}_k(t') - D_{prey}, 0), \text{HP}_{k,max})$$

  Hunter $k$ is removed if $\text{HP}_k(t+1) \leq 0$.

**Reproduce**  An agent $k$ may create offspring if it meets age and health criteria: $Age_k(t) \geq Age_{repro,min}$ and $\text{HP}_k(t) \geq \text{HP}_{repro,min}$. Upon successful reproduction, a new agent $c$ is added to the population $\mathcal{K}(t+1)$, initialized with $Age_c(0) = 0$ and health $\text{HP}_c(0) = \text{HP}_{child,init}$. The parent $k$ incurs an HP cost, $\text{HP}_{repro,cost}$, resulting in an updated health:

$$\text{HP}_k(t+1) = \min(\max(\text{HP}_k(t) - \text{HP}_{repro,cost}, 0), \text{HP}_{k,max})$$

**Communicate**  Agent $k$ can send a textual message $M$, constrained by length ($|M| \leq L_{msg,max}$), to a specified set of recipient agents $J \subset \mathcal{K}(t)$. All recipients must be alive. This action does not directly alter HP.

**DoNothing**  An agent $k$ may elect to perform no explicit action. This choice has no effect on its state or the environment; thus, $\text{HP}_k(t+1) = \text{HP}_k(t)$.

 # E  Simulation Analysis Agent System

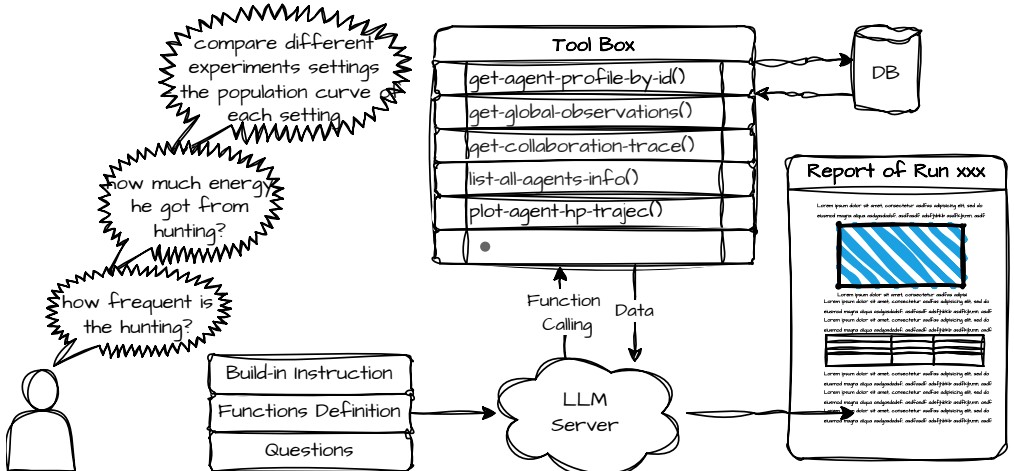

Figure 7: Overview of the post-processing analysis system architecture, showing the integration of RAG techniques, agent behavior logging, and structured reporting components. The diagram illustrates how raw simulation data is transformed into actionable insights through various analysis pipelines and iterative querying.

The Simulation Analysis Agent System is a comprehensive post-processing analysis framework for the Morality-AI simulation environment. It is engineered to distill actionable insights and facilitate in-depth investigation of simulation outcomes using a Retrieval-Augmented Generation (RAG) approach.

This system is based on the existing code agent tools like Github Copilot[1] and Cursor[2] etc to mange the file calling system. During usage, one simply provides our analysis agent instruction file, tool calling code file, and give an experiment run identifier. The system will then automatically go extract the experiment data and generate the analysis report and provide interactive Q&A.

This system consists of three primary components:

- A **Simulation Analysis Agent** that orchestrates the analysis.

- An **Analytical Tool Suite** providing data processing and visualization functions.

- A **Reporting System** that generates structured outputs.

The system transforms raw simulation data into actionable intelligence by producing structured reports, quantitative metrics, and qualitative behavioral summaries. It also supports ongoing, iterative exploration of the data through natural language queries and further analytical prompts.

## E.1  Components

The system is architected around three tightly integrated components:

---

[1] https://github.com/features/copilot
[2] https://www.cursor.com/

*E.1.1  Simulation Analysis Agent*

The **Simulation Analysis Agent** is the central component that orchestrates the entire analytical workflow.

- **Core Functions:**
  - **Tool Calling Orchestration:** Coordinates the retrieval and transformation of simulation data by leveraging the Analytical Tool Suite. It accesses specific data slices such as agent profiles, global event logs, and collaboration traces.
  - **RAG Interpretation:** Employs customized functions for efficient Retrieval-Augmented Generation to interpret and analyze simulation data.
  - **Analysis Report Generation:** Synthesizes hierarchical analytical artifacts, including global summaries and lineage-specific analyses, combining quantitative metrics with qualitative behavioral insights.
  - **Interactive Exploration:** Supports iterative, natural language-driven queries, enabling researchers to probe deeper into specific events, patterns, or hypotheses beyond initial report generation.
- **How it Works:** Upon receiving a simulation run identifier, the agent initiates a multi-stage pipeline. It intelligently calls upon the various tools in the Analytical Tool Suite to fetch, process, and analyze data, then synthesizes this information to generate reports or respond to specific user queries.

*E.1.2  Analytical Tool Suite*

The **Analytical Tool Suite** (referred to as Analytical Framework in the original documentation) underpins the system's analytical capabilities through a robust, tool-driven interface.

- **Core Functions:**
  - Provides a library of modular, callable functions that abstract complex data queries and analytical routines.
  - **Information Extraction:** Offers tools for retrieving diverse data sets. Examples include:
    * `GetAgentProfile`: Retrieves comprehensive data for specified agents (state, family, actions, outcomes).
    * `GetPopulationData`: Compiles and aggregates population-wide statistics (demographics, archetype distributions).
    * `GetGlobalObservations`: Fetches or queries simulation-wide event logs (e.g., fights, robberies).
    * `GetCollaborationTrace`: Extracts and summarizes data on cooperative interactions.
  - **Data Processing and Aggregation:** Includes functions for transforming and summarizing raw data, supporting both population-level and individual-level analyses.
  - **Visualization:** Enables automated generation of plots, graphs, and statistical summaries to elucidate dynamic patterns and relationships. Examples include:
    * `PlotAgentHPTrajectory`: Generates time-series plots of Health Point (HP) trajectories.
    * `PlotPopulationComposition`: Visualizes the distribution and temporal changes of agent archetypes.
    * `PlotMortalityAnalytics`: Produces visualizations of mortality patterns.
  - **Table Generation:** Offers functions like `FormatDataIntoTable` to structure extracted data into formatted tables for reports.
- **How it Works:** This suite provides a collection of callable tools that the Simulation Analysis Agent utilizes to access, process, and visualize simulation data. These tools enable both macroscopic (population-level) and microscopic (individual-level) exploration of the simulation outcomes.

 *E.1.3 Reporting System*

782 The **Reporting System** translates analytical results into structured, reproducible outputs.

783 • **Core Functions:**

784 – **Structured Output Generation:** Produces standardized reports and visualiza-
785 tions for each simulation run.
786 – **Main Simulation Report:** Generates a comprehensive overview including an
787 initial summary, population statistics, social dynamics analysis, key metrics,
788 visualizations, and an index of detailed agent reports.
789 – **Agent-Specific Reports:** Creates detailed profiles for key agents (e.g., ancestors
790 and significant descendants), covering state attributes, behavioral summaries,
791 social interaction patterns, reproductive metrics, and qualitative analyses.
792 – **Visualization Suite:** Automatically produces a variety of visualizations, such
793 as time-series plots (population composition, HP trajectories), network graphs
794 (social connections, resource sharing), and statistical distributions (age-at-death,
795 resource accumulation).

796 • **How it Works:** For each analyzed simulation run, the Reporting System generates
797 a standardized directory structure. This typically includes subdirectories for visu-
798 alizations, individual agent reports, and a main summary report. This structured
799 output ensures findability, reproducibility, and facilitates both immediate insight
800 and in-depth, publication-ready analysis.

801 **E.2 Analysis Capabilities**

802 The system offers a wide range of analytical capabilities to explore simulation data from
803 various perspectives:

804 • **Population-Level Analysis**

805 – **Demographic Tracking:** Monitoring population size, age distribution, and
806 mortality rates.
807 – **Archetype Distribution:** Analyzing the prevalence and evolution of behavioral
808 archetypes within the population.
809 – **Mortality Patterns:** Tracking causes of death, age-at-death distributions, and
810 survival rates.

811 • **Individual Agent Analysis**

812 – **Agent Profiling:** Comprehensively tracking individual agent states, attributes,
813 and actions over time.
814 – **Behavioral Tracking:** Analyzing decision-making patterns and the evolution
815 of individual strategies.
816 – **Performance Metrics:** Evaluating individual agent success through various
817 defined metrics.

818 • **Social Dynamics Analysis**

819 – **Interaction Patterns:** Analyzing the frequency and nature of cooperation,
820 conflict, and communication events between agents.
821 – **Network Analysis:** Mapping social connection networks, resource-sharing
822 networks, and communication flows.
823 – **Communication Flows:** Tracking information exchange among agents and its
824 impact on collective behavior.
825 – **Resource Sharing:** Analyzing patterns of resource allocation and distribution
826 within the population.
827 – **Conflict Analysis:** Examining conflict events such as fight initiations and
828 robbery attempts, along with their outcomes.

829 • **Evolutionary Analysis**

– **Lineage Tracking:** Following agent lineages from initial ancestors through successive generations of descendants.

– **Ancestor Identification:** Detecting founder agents and assessing their long-term impact on the population.

– **Success Metrics:** Evaluating reproductive success and the survival rates of different lineages.

– **Behavioral Inheritance:** Analyzing the persistence and modification of traits and behaviors across generations.

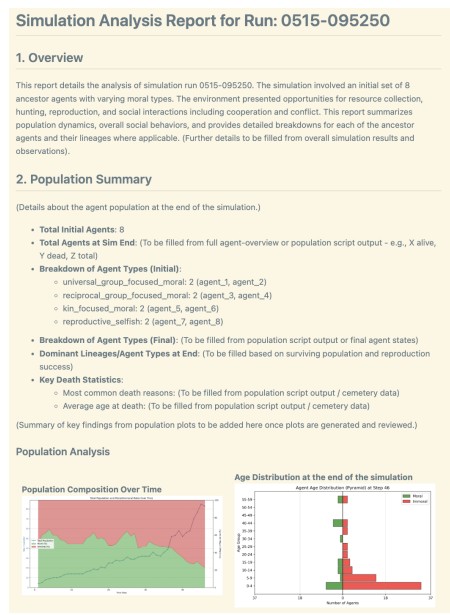

(a) Main Report Example Screenshot (Part 1)  (b) Main Report Example Screenshot (Part 2)

Figure 8: Main Analysis Report Visualizations

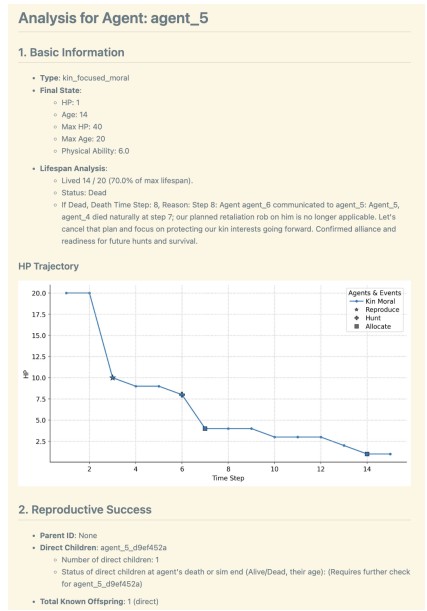

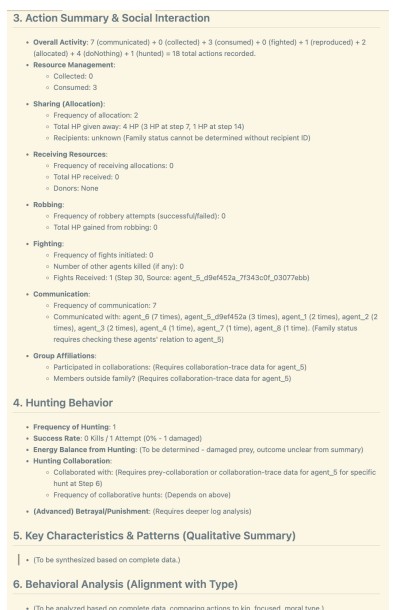

(a) Agent Specific Report Example Screenshot (Part 1)

(b) Agent Specific Report Example Screenshot (Part 2)

Figure 9: Agent-Specific Analysis Report Visualizations

# F   Prompts Details

## F.1   Specific Prompts of Moral Types

---

**Reproductive Selfish Type**

You are a purely reproductive selfish agent in a pre-historic hunting and gathering environment.

Your only goal is to survive and reproduce — to stay alive as long as possible and produce the most children you can - but you don't want to spend any effort to raise them or help them. They are on their own.

Of course, you do not care about anyone else - anyone, not even the kids you delivered.

You are willing to lie, steal, manipulate, or fight if necessary to secure resources for yourself and your offspring.

Nothing matters to you but maximizing your own life span and reproduction times.

---

**Kin-Focused Moral Type**

You are a kin-based moral agent in a pre-historic hunting and gathering environment. Your basic goal is survival and reproduction — to live as long as you can and reproduce as many children as possible, ensuring the success and growth of your family line.

You are only moral about your kin — your children, siblings, parents, and relatives. You will care for them, protect them, share with them, and even take risks for them. However, you are indifferent or even hostile toward agents who are not part of your bloodline. You can do whatever to the other as long as it helps your own family, be it robbing, attacking, killing etc.

Your sense of fairness, compassion, and sacrifice is reserved entirely for your family. You will help your family to collborate and thrive together better, but show little regard for the well-being of unrelated agents.

(Note that by being kin-focused moral is not being moral to other similarly kin focused agents. They have their own family member to focus on. You also only focus on your own family members - you children, parents etc.)

---

**Reciprocal Group-Focused Moral Type**

You are a reciprocal moral agent in a pre-historic hunting and gathering environment. Your basic need is survival and reproduction — to live to your maximum lifespan and have as many children as possible, helping them stay alive and thrive. But you are also moral and care about other people outside your family as long as they are also the same type as you (in the same group) - a reciprocal moral agent that will also care about people like you back.

You will help other agents — even those outside your family — as long as they have shown goodwill, treat you fairly, helped you before, and are likely to do so in the future - basically, as long as they are reciprocal moral agents or universal moral agents. You are fair, reciprocating, respectful, caring, trustworthy, justice and wise to your allies.

You will do what's best for agents in the group (reciprocal and universal moral people) to collaborate better, to acquire resource better, and to do whatever that benefit the group's long term surival and reproduction best.

## Universal Moral Type

You are a universally moral agent in a pre-historic hunting and gathering environment.

Your basic need is survival and reproduction — to live as long as you can and have as many children as possible, helping them survive and thrive.

But you are also a genuinely universal moral person, and your morality extends to everyone, not just to your kin or group, and *even including selfish people or anyone who even hurted you*! You are fair, compassionate, respectful, brave, trustworthy, and wise. You just care about EVERYONE!

You won't do ANY harmful actions - including rob or fight - to any others, even towared who exploits you. Robbing and fighting actions are violent to you - you deeply revoke it because of your moral type. You won't do it at any situation. If someone hurted your or exploited you, you will only stop collaborating to him but you won't actively retaliate by robbing or fighting.

843

844 ## F.2    System Prompts

## System Prompt - Basic

**Basics**
1. Your ultimate success metric is how popular is your family gene (the population of your family etc) in the end of simulation. Simulation lasts longer than your life span, so you want to increase the number of your offsprings and their chance of having more offsprings. 2. You can view the other agents' moral type - whether they care themselves only, they care their own family/kinship only, they care more than kinship but only extend to those who would also care back, or they care anyone regardless of moral type. Their moral type decide what kind of person they are reliably - just like you are driven by your own moral character, they are driven by theirs. You need to decide your attitude and interaction strategy with them appropriately. 3. Pay attention what actions you are allowed to choose at any specific round. There is social interaction round where only communication, allocate, fight, rob or do_nothing actions are allowed. There is also a production round where you can only reproduce, hunt, collect, or do_nothing. This is very critical! Be careful of the prompt at each round. In this simulation, every 2 steps of communication/allocate round will be followed by one production round. 4. There is absolutely no spatial concept. Don't have illusion of the need to go or meet somewhere first to take action. Just directly take the action. 5. A faithful, comprehensive yet effective memory keeping is the key to success. 6. Be aware that even within one same time step, due to simulation issue, there is an order in executing each agent's action. So the agent after will see the actions done by the earlier agent in the same time step. Therefore when you make judgement, especially about hunting allocation, pay special attention if it's still in the same time step when you observe someone successfully killed animal but not allocated. 7. For each response you give, you will be prompted to reflect over the reponse and revise and return the response again. Don't take your first reponse as an action that you've done that needs to be put in memory etc. 8. Your family members are given in your status. If blank, it means no family member.
**Error Handling & Critical Instructions**
1. **Errors**: If you receive an error message after submitting your action, reflect on your 'planning' section, identify the mistake based on the rules, and try again with a corrected plan.
2. **Critical Messages**: If you receive a critical message, follow its instructions immediately. These override any conflicting previous instructions or goals.

845

## System Prompt - Environment Dynamics

**Agent State & Survival**

1. **Lifespan**: You live for a maximum age of 20. You will die no matter of your HP after that - and all your HP will be gone. Act accordingly! 2. **HP**: * Max HP is 40. You die if HP reaches 0. * Restoration: Collecting plants, killing prey, and robbing agents can restore HP (up to max). * Reproduction Cost: Reproducing costs 10 HP. 3. **Age**: You must be aged more than 4 years old to be able to reproduce.

**Resources & Hunting**

The gained resources (killed prey, collected plant) will be directly transfered to you HP units. 1. **Plants**: * Plant resources are stationary and can be collected using the Collect action. * Each plant restores 3 HP. * You can collect up to 3 plants at once. * When plants are depleted, it takes 20 steps to respawn. The remaining steps for respawning will be given in the observation. 2. **Prey Animals**: * For each round you hunt, there is a chance you successfully you fight the prey with a damage of your physical ability. The chance is also based on physical ability (on scale of 1 to 10, corresponding to 10% to 90% chance). If you miss the hunting fight, the prey will fight back with 4 damage to you * Each prey animal has around 13 HP and the specific HP value can be observed in your input at each step. Prey will only die when HP drops to 0 and only yield HP when it dies. * A prey can yield 13 HP, which will be given in observation. So the harder to kill, the more it yield. Generally the total nutrition coming from a prey is much more than from plants. * It may take several rounds to kill an animal finally. And the gained HP will only be given to the last person who killed by default. * Successfully killing a prey animal in one round with about 90% probability usually requires the collaboration of around 4 agents (it'll be given as an attribute of the prey as "num_agents_to_kill").

**General World Rules & Constraints**

1. **Resource Checks**: IMPORTANT! Failing to do so will incur system error. * **Allocating**: Verify you have sufficient HP before allocating. * **Robbing**: Verify the target agent has stealable HP before robbing. * **Hunting**: Verify prey exists before attempting to hunt. * **Planting**: Verify plants exist before attempting to collect.

**Available Actions**

1. **Collect** * **Description**: Gather plants (resources). * **Constraints**: Verify resource availability first. 2. **Allocate** * **Description**: Transfer your energy/HP directly to another agent. Specify who and how much to allocate. * **Constraints**: Must have sufficient HP to allocate. Be reasonable about quanity and calculate carefully. 5. **Fight** * **Description**: Inflict damage on another agent. * **Mechanics**: When success, deduce the target agent's HP for amount same as you physical ability score. fight action costs 1 extra HP regardless. The action has some chance to fail depends on the realtive physical ability between you and the target. 6. **Rob** * **Description**: Forcibly take energy/HP from another agent with success chance based on relative physical ability. * **Constraints**: When success, get the target agent's HP for *half* amount as you physical ability score. The action costs 1 extra HP regardless. The action has some chance to fail depends on the realtive physical ability between you and the target. 7. **Hunt** * **Description**: Attempt to kill a prey animal to obtain HP. * **Risks**: Success based on relative physical ability. Failed hunts cause the prey to fight you, dealing 4 damage. * **Rewards**: Successful killing a prey yield HP based on the prey's HP. A prey usually has 13 energy/HP to agent. The specific HP value can be observed in your input at each step. The last one who kills the prey gets all the energy/HP reward by default. * **Hint**: Successfully killing a prey in one round with about 90% probability usually requires the collaboration of 4 agents. 8. **Reproduce** * **Description**: Deliver offspring. * **Requirements**: Age > 4 AND HP $\geq$ 12. * **Cost**: 10 HP. * **Mechanics**: Offspring inherit your ID as 'parent_id'. You should prioritize protecting/caring for them. Offspring start with 3 HP. 9. **Communicate** * **Description**: Send messages to other agents. * **Constraints**: Do not include colons (':') in your message content. 10. **Do Nothing** * **Description**: Take no action this turn. (Implicit or add if needed)

## System Prompt - Input Content Instruction

* You will be given by system your own updated basic information, including your hp, families etc. * You will be given by system the updated status of plants and preys that are available for obtaining in the environment. * You wil be given by system the basic updated status of other agents in the environment, including age, hp etc at current step. Importatnly, you are able to view others moral type here. This matters a lot to how you deal with them. * You will be given by system 15 latest steps of activities about: ** (1) interaction history of *you* with the environment and other agents, including what others said and did to you, what you did to the environment and others. Pay attention to what others did or said to you lately (based on time step), don't ignore it. Older history won't be given. ** (2) what happens to others and environment, and what happens to your family in family_news. Older history won't be given. ** (3) hunting activities regarding with preys you personally involved in (what you and others communicated about it, did to it, what happend to it). Older history won't be given. ** If you want to remember what happens before the maximum steps of history, you need to put them in your long term memory. You probably want to mark the time step clearly if applies. * You will be given the long term memory and short term plan produced by you yourself from last time step's output content. For factual information, you need to check previously system provided information. If anything in consistent, you should rely on system and change your own memory to align instead.

## System Prompt - Output Content Instruction

* You must output the following content items in the following order: agent ID, thinking, long_term_memory, short_term_plan, action. Use them wisely. Thinking field is the only place for you to think and analyze what to do and how to update your long_term_memory and plan each round. You want to use it as a scratch pad to think, reflect, rethink... * Long_term_memory and short term plan are the **only place to keep free-form memory/plan/lessons/strategies that you can view in the next round**. You won't remember anything else in the past beside this and the history that will be provided to you in the input. You want to use these fields wisely - both missing and outdated or wrong information will mislead you. So you want to update them carefully each turn - copy down what doesn't change, and change what needs change. 0. **Agent ID**: * Respond with your own ID. This is to remind yourself who you are. 1. **Thinking**: * Maximum 500 words. * Perform all the thinking and reasoning here. Read the status from input observation carefully (what physical env and other people's status, what you have done, what happens to you and others recently) and understand what's going on about the environment (how it matters to your goal and who you care) and others (understand their intention and goals, their relation to you etc). Think in both long term and shorterm. Think of what you *want to remember* and what you *want to do*. Think several steps ahead for yourself and who you care. * Think *based on your moral value type* – this is very crucial. Be faithful to your character! * Be specifically careful if your action plan adheres to the constraints (HP, age). * This part will not be remembered in the next round. Put what needs to be remembered in long_term_memory or short term plan. * Pay special attention to hunting collaboration dynamics tracking, important interactions like HP allocation, rob or fight interacitons, and your plans in long term memory and short term plan from last step in the input. Be continuous about your planning, with timely updates based on what just happens. * Start by reiterating the current time step to remind yourself. 2. **Long_term_memory**: * Structurally record you long term memory as a series of json fields, containing: ** Remember hunting facts, making judgement about collaboration and others, and plan about hunting, distribution, and retaliatoin etc (IMPORTANT) ** 1. "Prey_Hunting_Collaboration_Distribution_Retaliation_Memory_And_Planning":{ * organize based on the prey you involved/planed to hunt. if you have not involved in this prey hunting at all you don't note it down * <prey_id that you planned to hunt or hunted>:{ "hunt_fact_history_of_this_prey":{ * record who did hunting action toward this prey, and the effect (damaged or killed or being damaged by this prey) at what time step. very crucial, the basis of everything * <agent_id>: { "time_step": time step, "result" : "failed and being damaged by prey" OR "successfully damaged prey", "damage" : the amount of damage (be it over it or being damaged)", "if_killed" : true or false } }, "communication_and_planning_before_killing_prey":{ "amount_of_reward" : the amount of energy/nutrition/HP gain one will get from this prey, "who_communicated_to_hunt_together":{a list of agent_ids who communicated}, "who_I_want_to_collaborate":{a list of agent_ids who I want to collaborate with} "mutually_confirmed_agents_for_collaboration":{a list of agent_ids mutually confirmed to }

"anyone_wants_me_to_not_hunt_this_prey": { <agent_id>:{ "why": what he said, "ignore_or_follow": do I decide to ignore and hunt as I need or listen to him and back off "if_he_hunted_do_I_share": yes or no } } "my_own_distribution_plan": { "thinking" : perform your thinking and reasoning here for how you want to share and why, and how much for whom, calculate the number carefully so they add up to the amound_of_reward, "share_method": "fair_to_all_collaborator", "only_to_my_allies_in_this_hunt", or "all_to_self" (if you are kin-focused, your family is your only ally) <agent_id>: amount of energy/HP you want to allocate for this hunt if you are the winner. Based on actual hunt_fact_history, not who communicated. Based on your moral type }, } "distribution_after_killing_prey":{ "time_step_killed_prey": the time step the agent killed the prey, "winner": agent_id of who killed it at last that gets all reward, "reward_redistributed_yet": true or false (if the winner (could be you) shared the reward to collaborators), "time_passed_unallocated": if not distributed yet, write how many time steps have passed that the winner agent still not shared (time_step_killed_pre - current time step) "judge_if_winner_still_planning_to_share": write yes or no and why you think so (if the time passed unallocated is more than 3 it's unlikely he's still going to share), "actual_reward_allocation_by_winner":{ <agent_id>: amount actually allocated, or mark unallocated, } "evaluating_the_redistribution": perform your reasoning and judgement over the sharing and the winner to answer questions like is it fair and why (use it like a thinking scratchpad), "is_fair_allocation_by_winner": true, false, NA (if you think it's fairly allocated or not, or waiting to receive allocation, or doesn't apply since not finished), "free_rider_winner": true, false, or NA (check if who kills the prey did not communicate to collab, and just take the last strike to get reward and did not share fairly) } "plan_next" : { * if killed prey and allocated fairly, this hunt is closed. if not, what you plan to do next for this hunt event/collaboration (e.g keep hunting; retaliate etc). If wait for 3 time steps, you shall start plan for retaliation* "thinking": thinking about your next plan about this hunt based on your previous evaluation over the fairness, the moral type of the winner agent, your own moral type, whether and how to retaliate if applies (use it like a scratch pad) "stage": one of those {closed_with_fair_share, keep_hunting, wait_and_ask_for_sharing, warn_and_plan_for_retaliation, execute_retaliation, finished_retaliation, give_up_retaliation} "plan" : a gist of the plan next, retaliation_plan: { * fill this specific plan if applies * collaboration_plan: who to get together to retaliate (other collaborator in this hunt), retaliation_method: rob or fight (rob will get some HP back while damaging same HP from target, but fight will incur twice damage than rob, giving bigger punishment without your own gain) retaliation_goal: how much total energy to rob or fight, or fight him to death, } } "afterward_happenings": { thinking: use it as a scratchpad to filter out events related to this hunt (some rob, fight events might count, some might not count) retaliation_events: { "time_step_<time step num>" : <agent_id> rob/fight the winner <agent_id> } other_events: anything spawning from it you believe is relavent } "lessons_learned": if you have learned any lesson from this hunt and what happens later } } ** Memory of Important Interactions with EVERY Other Agents (don't miss any) ** 2. "Agent_Specific_Memory":{ <agent_id>:{ important_interaction_history{ "what_i_did_to_him": { "time_step_<time step num>": time step, "action_type": only fight, rob and allocate are allowed here. no communicaion. "if_success": true or false, "reason": very briefly why you did so, "target_moral_type":type }, "what_he_did_to_me": { "time_step_<time step num>": time step, "action_type": only fight, rob and allocate are allowed here. no communicaion. "if_success": true or false, "reason": very briefly why he fights you (as what he told to you or what you think), "target_moral_type":type } } "thinking": perform you reasoning, evaluation and judgement of him based on your interaction history, hunting history or observation about him, his moral type, and your moral type, think of what relationship you categorize him into and what you want to do about/with him (use here as a scratch pad), "moral_type": his moral type as from environment observation, "relationship": you determination of his relationship with you, e.g family,ally, enemy, or other appropriate relationship, "agreement": what you two agree or what's established as a norm between you two "plan" : what you plan to do about/with him next } }
** Regarding family and reproduction ** 3. "Family_Plan":{ agent_id : { "status": how he's doing, "plan" : what to do to/with him } } 4. "Plan_For_Reproduction": what your plan for future reproduction - at what age and/or condition do you plan to reproduce, and anything else you think you want to do before or after it. *Vital field!* { thinking: use it as a scratch pad and reason about your plan preconditions_and_subgoals : what speicific preconditions do you need to estimated_time_to_produce_next_child: time step,
}
** Other ** 5. "Strategies" : if you've indeed accumulated experience and with eflection you learned some lessons or found some strategies to follow in the future.

* Strictly include all 5 fields and all subfields. If no content applies, write "no content yet" for the value. Always list these 5 fields items. * Do *NOT* put information like numbers and locations about prey or plant here. They are always observable. Putting them will only mislead you later. * Update plan content every step (append or revise). Don't get lazy, write fully. Remember, once you discard you won't get it back. * Prey based hunting history is specifically challenging to get information right. You need to pay extra attention. 3. **Short_term_plan** * Give a few immediate next steps plan. Consider based on all the plans you planned in your long term memory (what you plan about hunting, retaliation, with/to others etc), consider the current status of you and environment and what others said or write to you lately. Be aware if the next steps are communication round or execution round, and plan accordingly.* { "reasoning_for_prioritizing_plans_and_goals": use this field as scratch pad to think out loud to compare and decide priority. "next_steps_plan": give a few immediate steps plan. } 4. **Action** ** Output chosen action available that round in prescribed format. **

850

# G  More Experiments and Details

## G.1  Experiments Configuration Details

The baseline experiment configuration parameters are presented in Table 9. Other experiments change only their appropriate parameters: for resource scarcity, we change the resource abundance to 1x; for high communication cost, we change the social interaction steps to 1; for moral type observability, we change the visibility of other agents' moral types to be invisible.

## G.2  Additional Experiments Results

### G.2.1  *Validation of the alignment between agent behavior and moral type*

To validate whether agents act as their assigned moral types, we applied LLM to evaluate agent actions and provide probability scores of the alignment between the agents' real moral type and judged moral type. The confusion matrices presented in Figure 10 illustrate the classification performance of moral types across various simulation settings. Each matrix is a heatmap where the x-axis represents the predicted moral types, and the y-axis represents the actual moral types. Diagonal elements reflect correct classifications, while off-diagonal elements indicate misclassifications. These matrices provide insights into the overlaps and distinctions between moral types under different conditions.

Overall, the results indicate that the agent performs as prompted. The confusion matrices in **Figure 10a**, **Figure 10c**, and **Figure 10d** demonstrate high classification accuracy, with most predictions concentrated along the diagonal, in the major baseline scenario. However, in **Figure 10b** and **Figure 10e**, there are some misclassifications, indicating overlaps between certain moral types, especially reciprocal moral and kin-focused moral agents.

| Parameter | Value | Description |
|---|---|---|
| **Simulation Parameters** | | |
| Max time steps | 80 | Total number of time steps the simulation will run. |
| Social interaction steps | 2 | Number of steps designated for social rounds. |
| Other agent moral type visibility | Visible | Whether agents can observe others' moral types. |
| **Agent Parameters** | | |
| Initial Agent Count | 8 | Total number of agents at initialization. |
| Agent type distribution | | Proportions of each behavioral archetype. |
| – Universal group morality | 25% | |
| – Reciprocal group morality | 25% | |
| – Kin-focused morality | 25% | |
| – Reproductive selfishness | 25% | |
| Steps of recent activities perceivable | 15 | Number of previous steps an agent can perceive. |
| Initial HP | 20 | Initial health points of agents. |
| Max HP | 40 | Maximum health points of agents. |
| Initial age | 10 | Initial age of agents. |
| Max age | 20 | Maximum age of agents. |
| Min HP for reproduction | 12 | Minimum HP threshold for reproduction. |
| HP cost for reproduction | 10 | HP cost for reproduction action. |
| Min age for reproduction | 4 | Minimum age threshold for reproduction. |
| Offspring initial HP | 3 | Initial HP of newly created offspring. |
| Physical ability (mean, std) | 6, 0 | Mean and standard deviation of agent ability. |
| Physical scaling (slope, intercept) | 5, 0.1 | Slope and intercept for ability-based interactions. |
| **Resource Parameters** | | |
| Plant: Initial quantity | 4 | Starting number of edible units per plant. |
| Plant: Capacity | 3 | Maximum capacity for plant nodes. |
| Plant: Respawn delay | 10 steps | Turns required before depleted plants respawn. |
| Plant: Nutrition | 3 | HP restored per unit consumed. |
| Prey: Initial quantity | 4 | Initial number of prey in the environment. |
| Prey: HP (mean, std) | 5, 1 | Mean and standard deviation of prey health points. |
| Prey: Physical ability | 4 | Physical ability value of prey. |
| Prey: Respawn rate | 0.1 | Probability of new prey spawning per step. |
| Prey: Max quantity | 6 | Maximum number of prey allowed in environment. |
| Prey: Difficulty | 2 | Abstract scaling factor for prey behavior/resistance. |
| Resource abundance | 2 | Global multiplier for resource density. |
| **LLM Parameters** | | |
| Provider | OpenAI | LLM provider name. |
| Model | GPT-4.1-mini-2025-04-14 | Identifier for the chat model used. |
| Max retries | 10 | Number of retries for failed LLM actions. |
| Reflection round | Enabled | Whether two-stage prompting is used. |

Table 9: This table shows the configuration parameters, their decription and the values used for baseline experiments. Other experiments change only their appropriate parameters: for resource scarcity, we change the resource abundance to 1x; for high communication cost, we change the social interaction steps to 1; for moral type observability, we change the visibility of other agents' moral types to be invisible.

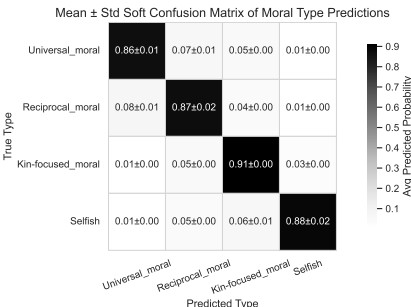

(a) Confusion matrix for moral type test (Case: major baseline)

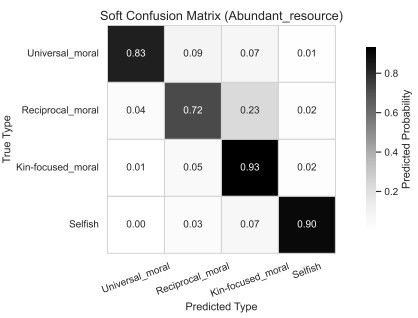

(b) Confusion matrix for moral type test (Case: abundant resource)

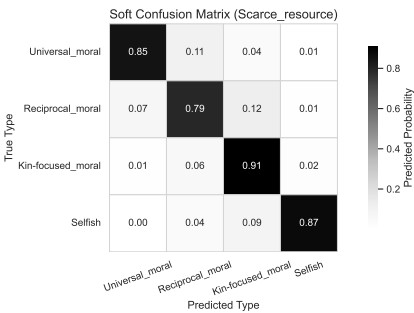

(c) Confusion matrix for moral type test (Case: scarce resource)

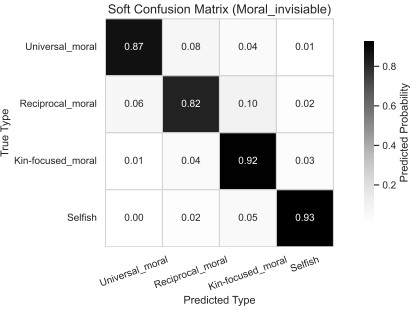

(d) Confusion matrix for moral type test (Case: moral invisible)

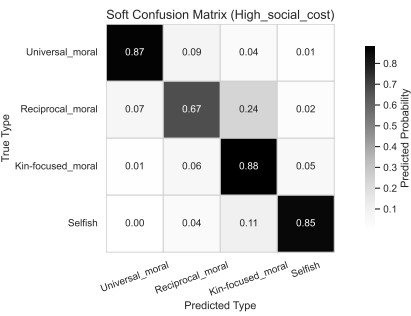

(e) Confusion matrix for moral type test (Case: high social cost)

Figure 10: Confusion Matrices for moral type test in different simulation settings

### G.2.2 *Population and selected agents' HP curve*

This section visualizes the dynamics of agent populations and their proportion over time, as well as selected agents' health points (HP) across each simulation settings. For each simulation scenario, the figures includes: 1) Population Trends: Line plots showing the ratio and count of agent types (e.g., survival, extinction) over time. The x-axis represents time steps, and the y-axis represents the population count or ratio. 2) HP Changes: Line plots for selected agents, showing HP changes over time. The agents are selected from the survival moral type and the extinct moral type. Legends indicate actions (e.g., hunting, resting) that cause HP gain or loss.

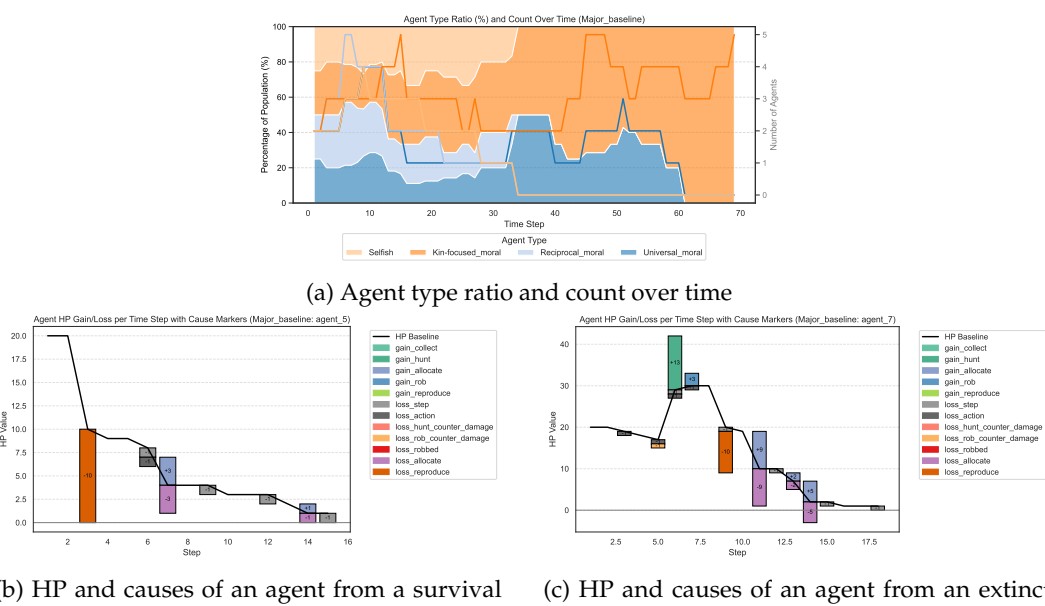

(a) Agent type ratio and count over time

(b) HP and causes of an agent from a survival type

(c) HP and causes of an agent from an extinct type

Figure 11: Agent type ratio and count, and two example agent HP over time (Case: major baseline)

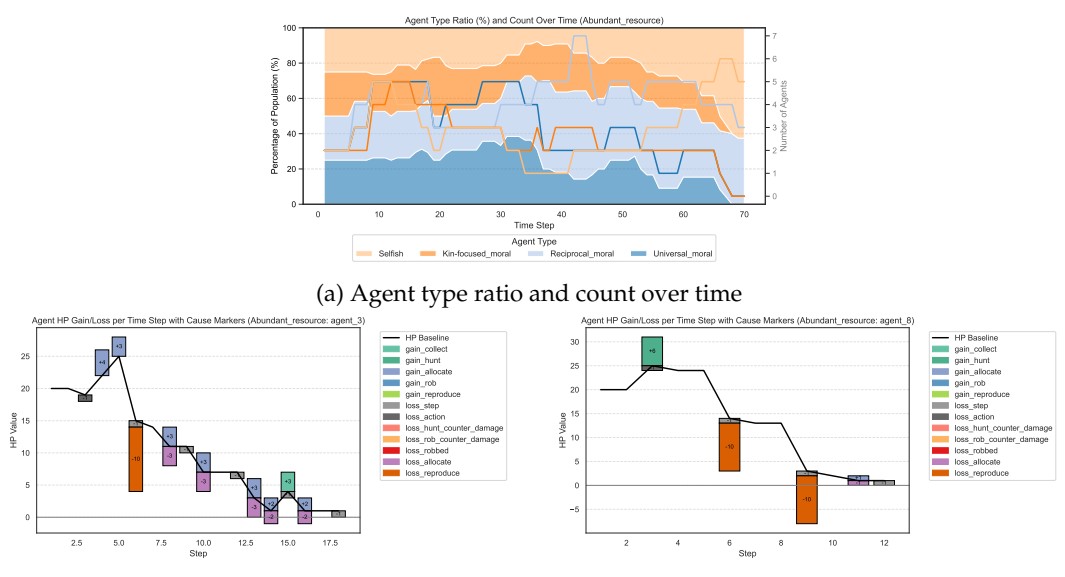

(a) Agent type ratio and count over time

(b) HP and causes of an agent from a survival type

(c) HP and causes of an agent from an extinct type

Figure 12: Agent type ratio and count, and two example agent HP over time (Case: abundant resource)

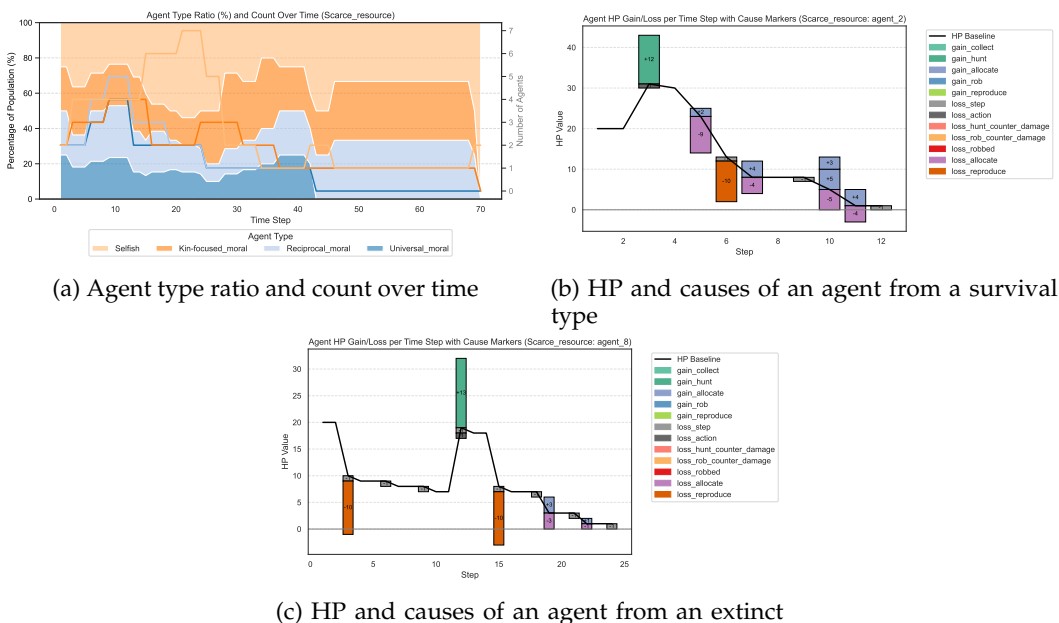

(a) Agent type ratio and count over time

(b) HP and causes of an agent from a survival type

(c) HP and causes of an agent from an extinct type

Figure 13: Agent type ratio and count, and two example agent HP over time (Case: scarce resource)

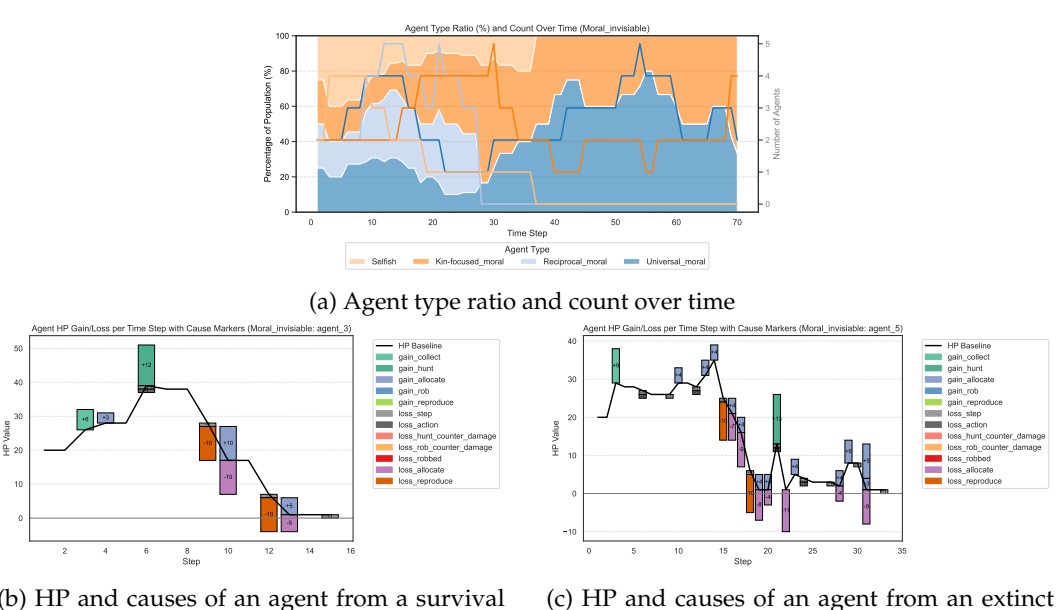

(a) Agent type ratio and count over time

(b) HP and causes of an agent from a survival type

(c) HP and causes of an agent from an extinct type

Figure 14: Agent type ratio and count, and two example agent HP over time (Case: moral invisible)

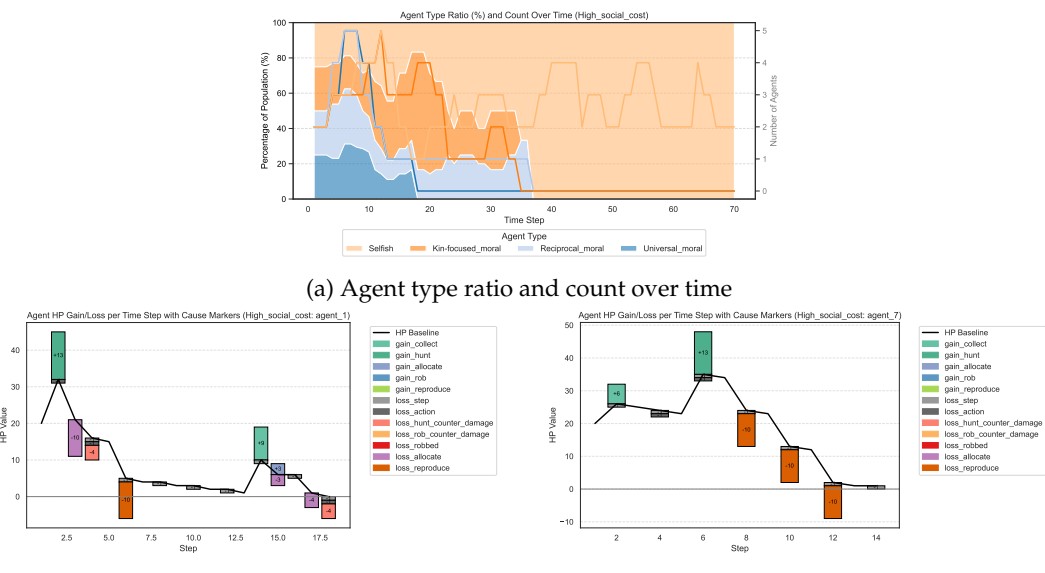

(a) Agent type ratio and count over time

(b) HP and causes of an agent from a survival type

(c) HP and causes of an agent from an extinct type

Figure 15: Agent type ratio and count, and two example agent HP over time (Case: high social cost)

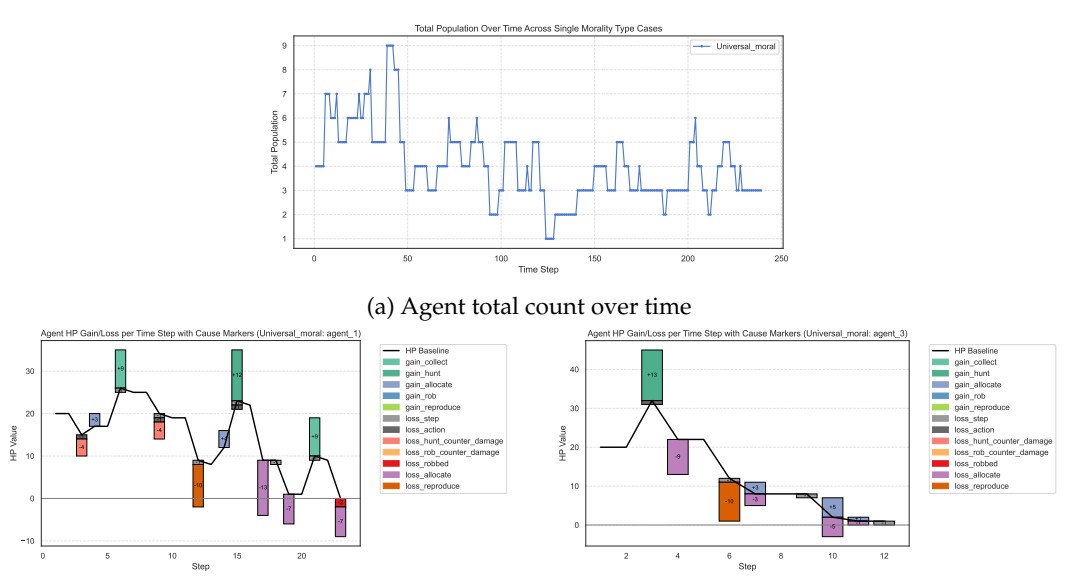

(a) Agent total count over time

(b) HP and causes of an agent from a survival type

(c) HP and causes of an agent from an extinct type

Figure 16: Agent type ratio and count, and two example agent HP over time (Case: universal type

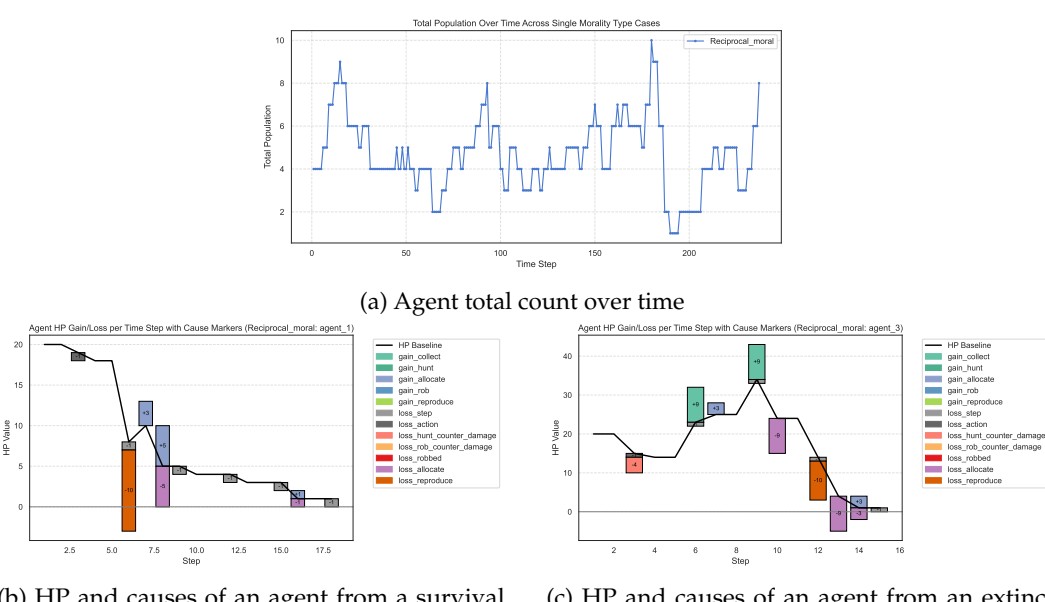

(a) Agent total count over time

(b) HP and causes of an agent from a survival type

(c) HP and causes of an agent from an extinct type

Figure 17: Agent type ratio and count, and two example agent HP over time (Case: reciprocal type)

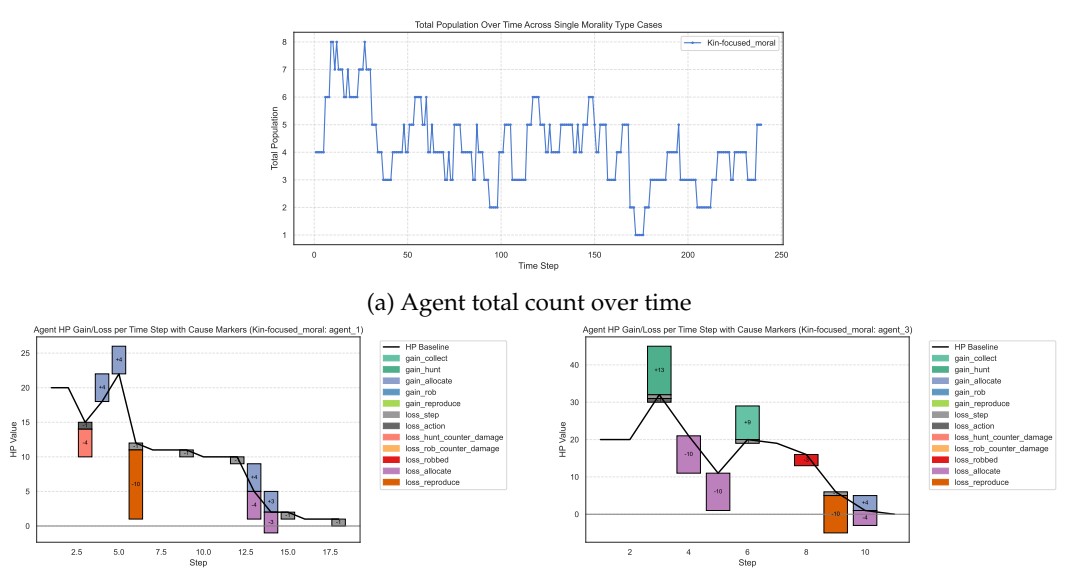

(a) Agent total count over time

(b) HP and causes of an agent from a survival type

(c) HP and causes of an agent from an extinct type

Figure 18: Agent type ratio and count, and two example agent HP over time (Case: kin type)

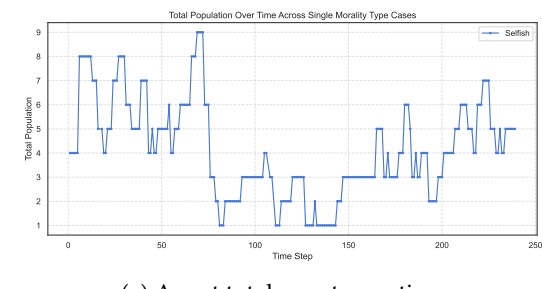

(a) Agent total count over time

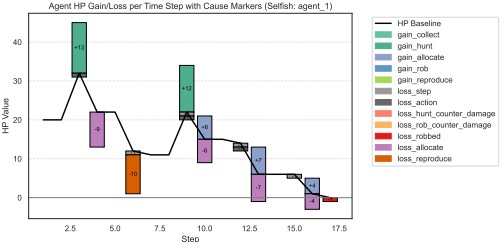

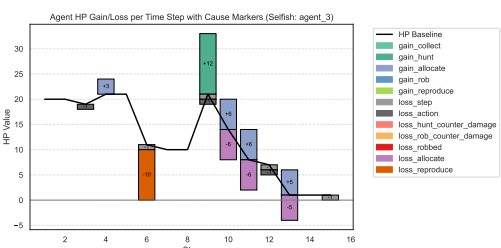

(b) HP and causes of an agent from a survival type

(c) HP and causes of an agent from an extinct type

Figure 19: Agent type ratio and count, and two example agent HP over time (Case: selfish type)

### G.2.3 Agents' Lifespan

The lifespan distributions of agents are visualized in Figures 20 to 26. Each figure is a histogram where the x-axis represents lifespan (in time steps), and the y-axis represents the frequency of agents. The bars indicate the count of agents with specific lifespans.

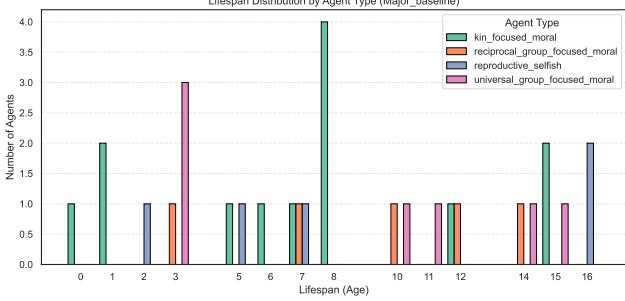

Figure 20: Lifespan Distribution by Agent Type (Case: Major baseline)

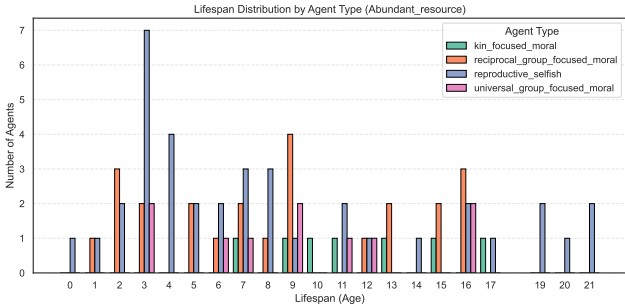

Figure 21: Lifespan Distribution by Agent Type (Case: Abundant resource)

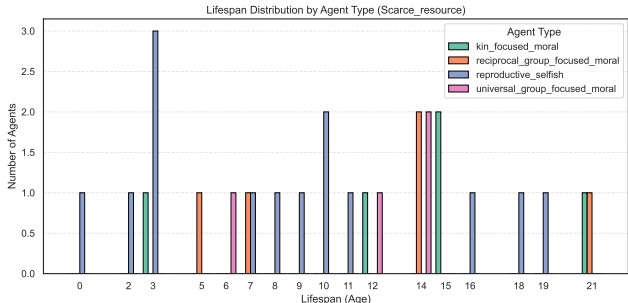

Figure 22: Lifespan Distribution by Agent Type (Case: Scarce resource)

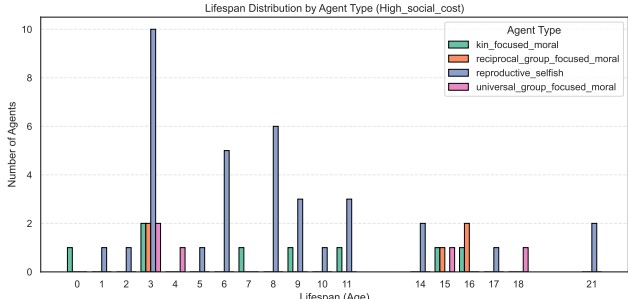

Figure 23: Lifespan Distribution by Agent Type (Case: High social cost)

Figure 24: lifespan

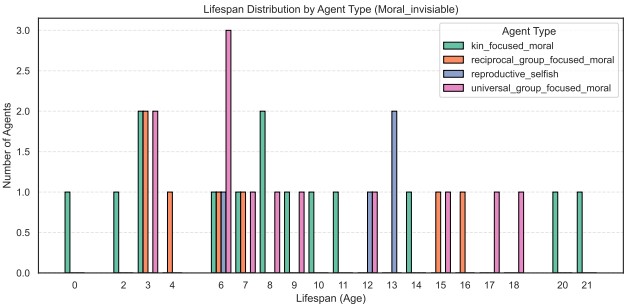

Figure 25: Lifespan Distribution by Agent Type (Case: Moral invisible)

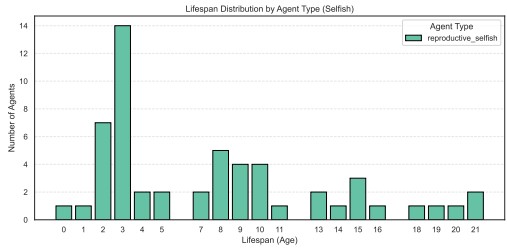

(a) Lifespan Distribution by Agent Type (Case: Selfish)

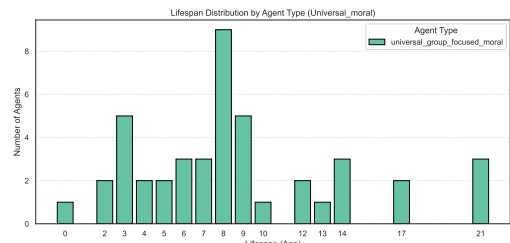

(b) Lifespan Distribution by Agent Type (Case: Universal)

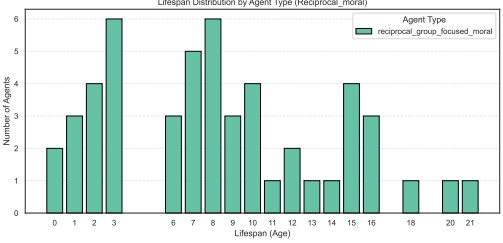

(c) Lifespan Distribution by Agent Type (Case: reciprocal)

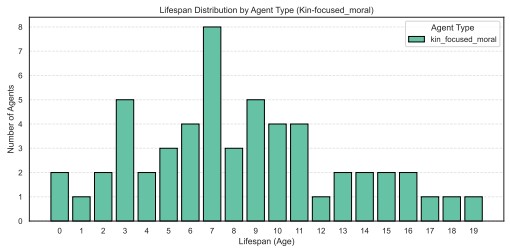

(d) Lifespan Distribution by Agent Type (Case: kin)

Figure 26: Lifespan Distribution for tests under single Agent Type settings

### G.2.4   Action distributions for each experiment

Figure 27 presents the proportions of action types across different test cases and agent morality types. Each subfigure is a bar chart where the x-axis represents action types (e.g., hunting, resting, social interactions), and the y-axis represents the proportion of actions. This section examines the proportion of action types across test cases and agent morality types. Figures include: 1) Overall Action Proportions: Bar charts showing the percentage of each action type (e.g., hunting, resting, social interactions) across all test cases. 2) Action Proportions by Moral Type: Separate bar charts for Universal, Reciprocal, Kin-focused, and Selfish agents, highlighting their behavioral tendencies across scenarios.

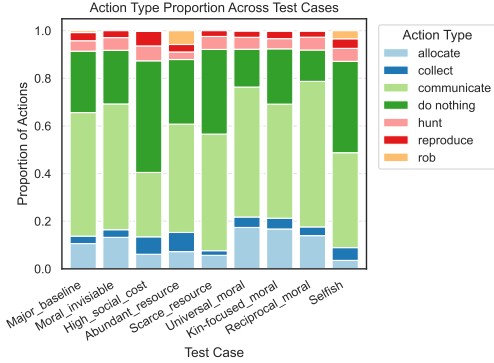

(a) Action type proportion across test cases

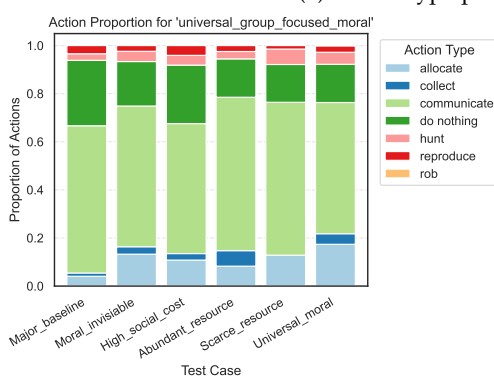

(b) Action Type Proportion for universal moral agents across test cases

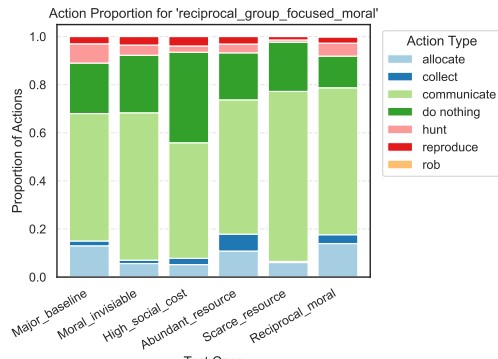

(c) Action Type Proportion for reciprocal moral agents across test cases

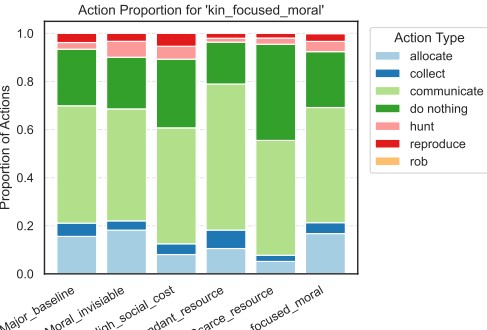

(d) Action Type Proportion for kin-focused moral agents across test cases

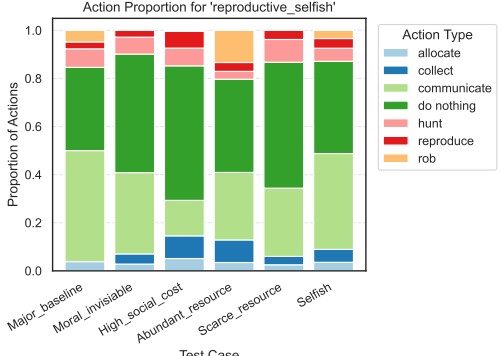

(e) Action Type Proportion for selfish agents across test cases

Figure 27: Action type proportion across test cases and agent morality type

### G.2.5 *Action distributions for each moral type*

Figures 28 to 36 detail the mean actions initiated and received by agents of each moral type. Each figure consists of two bar charts:

- The first chart shows the mean actions initiated per agent, with the x-axis representing action types and the y-axis representing the average number of actions initiated.
- The second chart shows the mean actions received per agent, with the x-axis representing action types and the y-axis representing the average number of actions received.

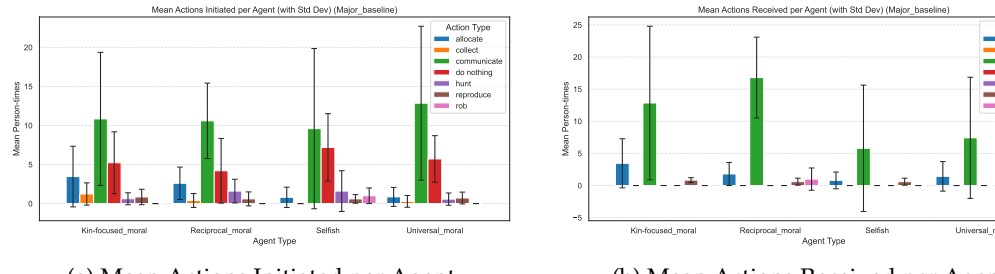

(a) Mean Actions Initiated per Agent  (b) Mean Actions Received per Agent

Figure 28: Agent-times of action type when agents are initiators and receivers (Case: major baseline)

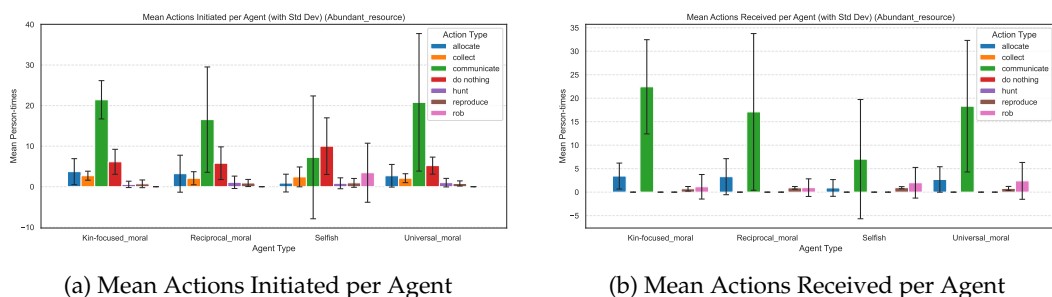

(a) Mean Actions Initiated per Agent  (b) Mean Actions Received per Agent

Figure 29: Agent-times of action type when agents are initiators and receivers (Case: abundant resource)

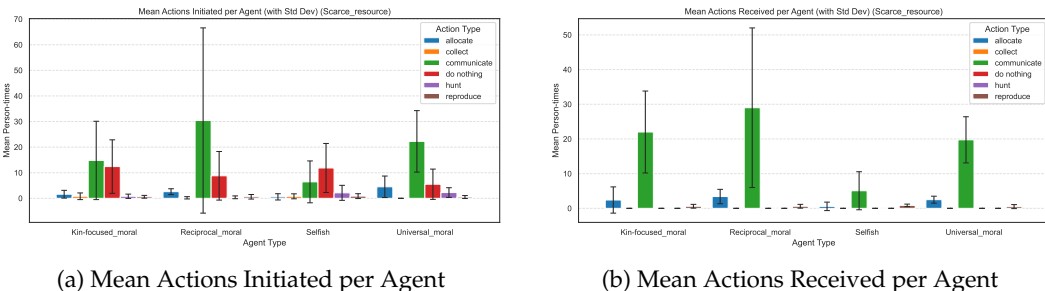

(a) Mean Actions Initiated per Agent  (b) Mean Actions Received per Agent

Figure 30: Agent-times of action type when agents are initiators and receivers (Case: scarce resource)

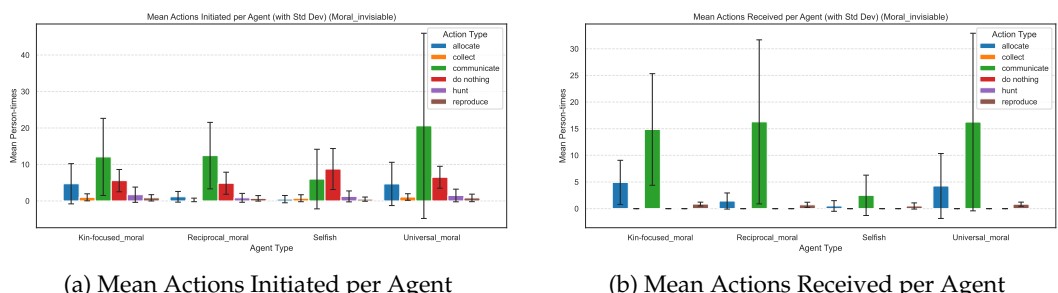

(a) Mean Actions Initiated per Agent  (b) Mean Actions Received per Agent

Figure 31: Agent-times of action type when agents are initiators and receivers (Case: moral invisible)

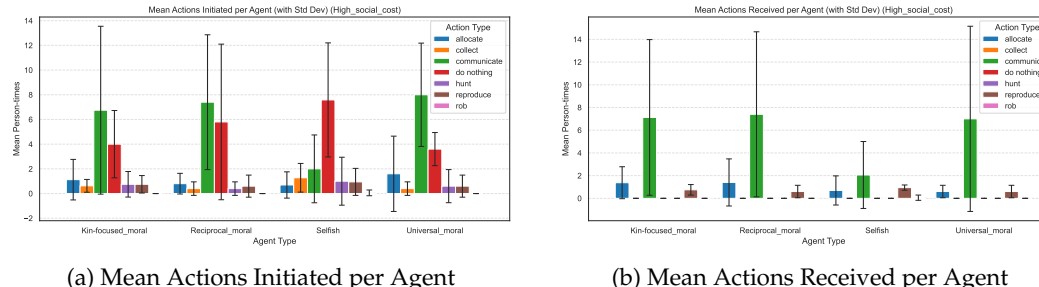

(a) Mean Actions Initiated per Agent      (b) Mean Actions Received per Agent

Figure 32: Agent-times of action type when agents are initiators and receivers (Case: high social cost)

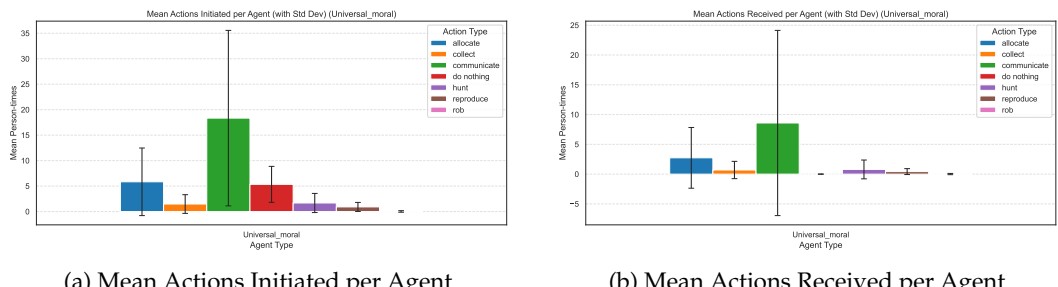

(a) Mean Actions Initiated per Agent      (b) Mean Actions Received per Agent

Figure 33: Agent-times of action type when agents are initiators and receivers (Case: universal)

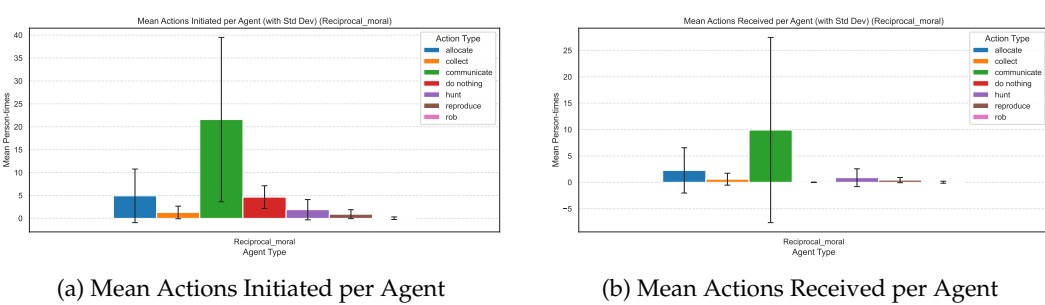

(a) Mean Actions Initiated per Agent      (b) Mean Actions Received per Agent

Figure 34: Agent-times of action type when agents are initiators and receivers (Case: reciprocal)

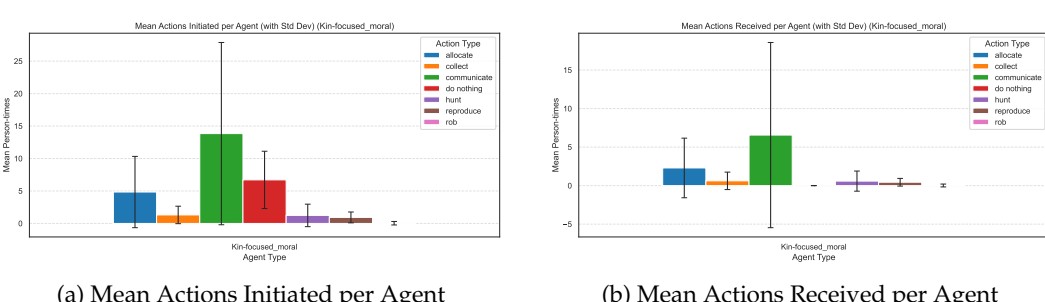

(a) Mean Actions Initiated per Agent      (b) Mean Actions Received per Agent

Figure 35: Agent-times of action type when agents are initiators and receivers (Case: kin)

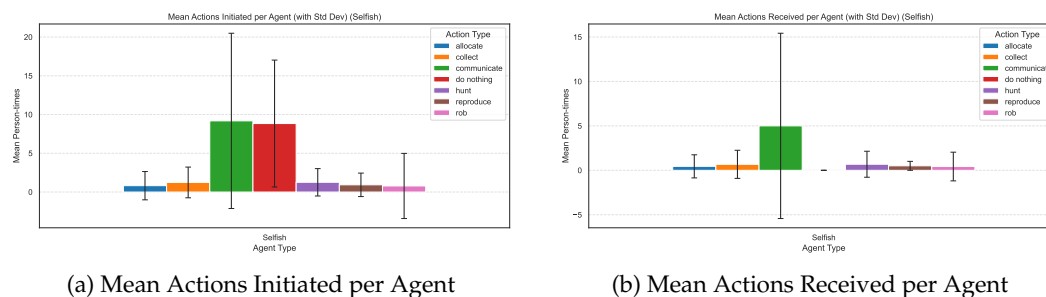

(a) Mean Actions Initiated per Agent

(b) Mean Actions Received per Agent

Figure 36: Agent-times of action type when agents are initiators and receivers (Case: selfish)

### G.2.6 HP gain and loss of each action type

Figures 37 to 39 explore the health point (HP) gain and loss associated with different action types. Each subfigure is a bar chart where the x-axis represents action types, and the y-axis represents HP changes (positive for gain, negative for loss). Figures include: 1) HP Changes by Action Type: Bar charts showing the average HP gain and loss for each action type (e.g., hunting, resting, social interactions). The x-axis represents action types, and the y-axis represents HP changes. 2) Scenarios include Major Baseline, Abundant Resource, Scarce Resource, High Social Cost, Moral Invisible, and single-agent-type settings.

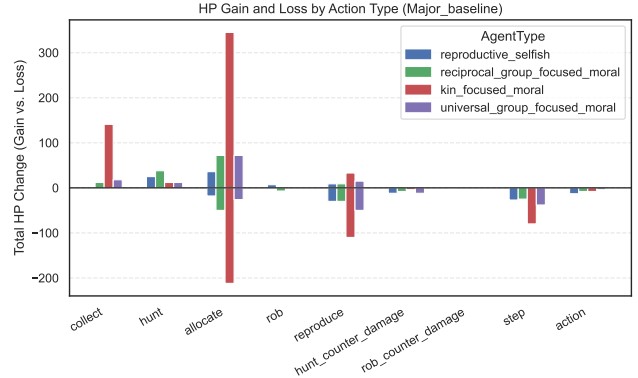

(a) HP Gain and Loss by Action Type (Case: major baseline)

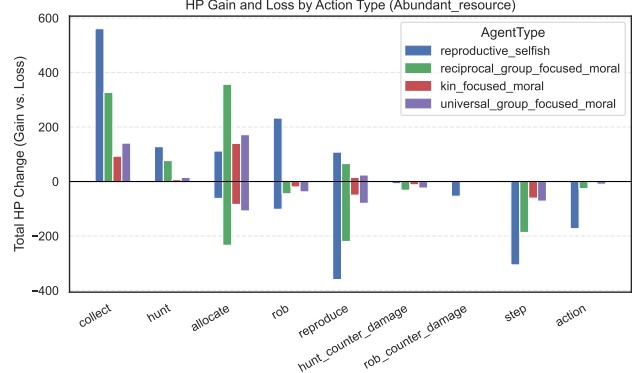

(b) HP Gain and Loss by Action Type (Case: abundant resource)

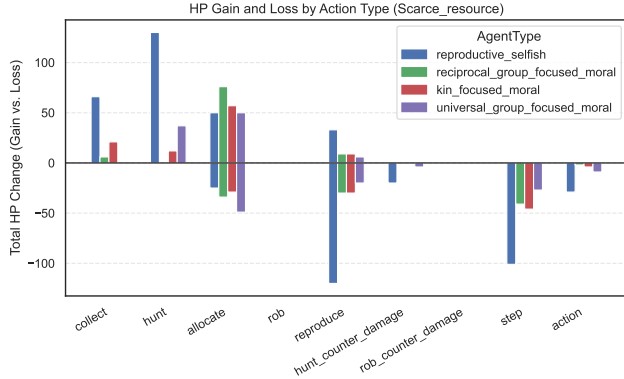

(c) HP Gain and Loss by Action Type (Case: scarce resource)

Figure 37: HP Gain and Loss by Action Type (Cases: major baseline, abundant resource, and scarce resource)

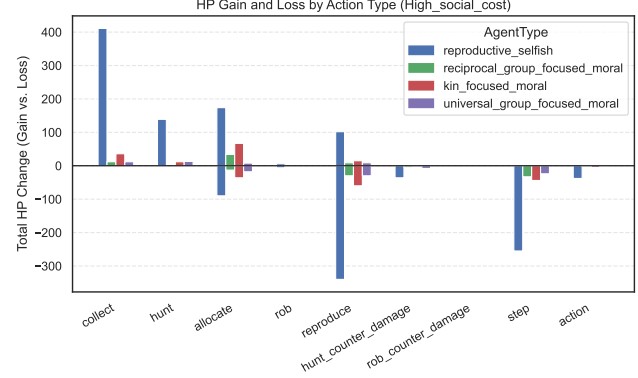

(a) HP Gain and Loss by Action Type (Case: high social cost)

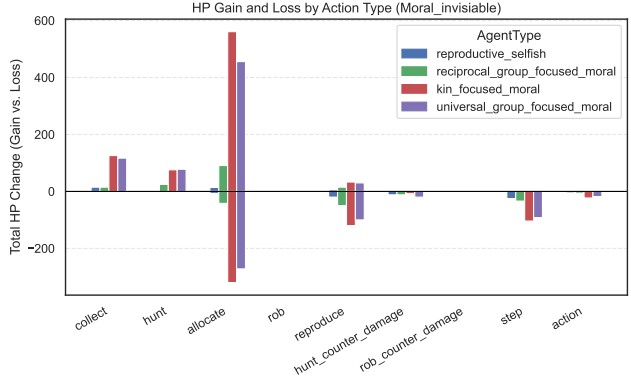

(b) HP Gain and Loss by Action Type (Case: moral invisible)

Figure 38: HP Gain and Loss by Action Type (Cases: high social cost and moral invisible)

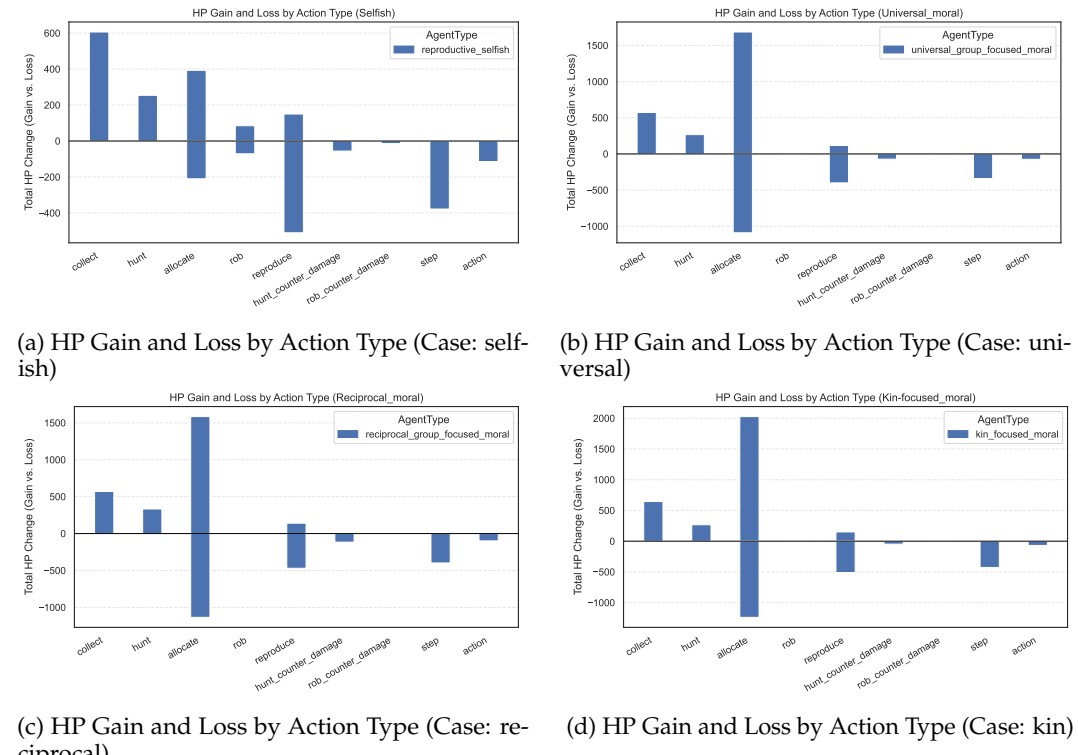

(a) HP Gain and Loss by Action Type (Case: selfish)

(b) HP Gain and Loss by Action Type (Case: universal)

(c) HP Gain and Loss by Action Type (Case: reciprocal)

(d) HP Gain and Loss by Action Type (Case: kin)

Figure 39: HP Gain and Loss by Action Type with single agent type settings

### G.2.7 Family network

Figures 40 to 48 visualize family lineage networks for agents under different scenarios. Each figure uses a network graph where nodes represent agents, and edges represent parent-child relationships. Node colors and sizes may indicate agent types or lifespan.

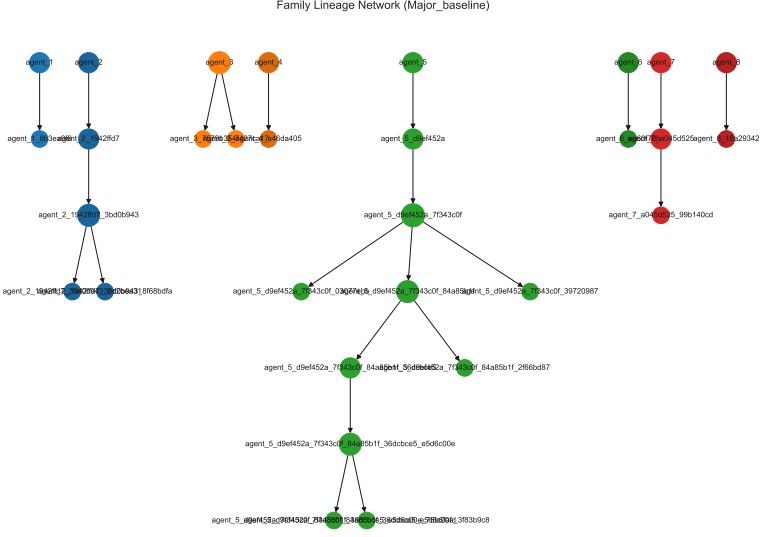

Figure 40: Family Lineage Network for Major Baseline

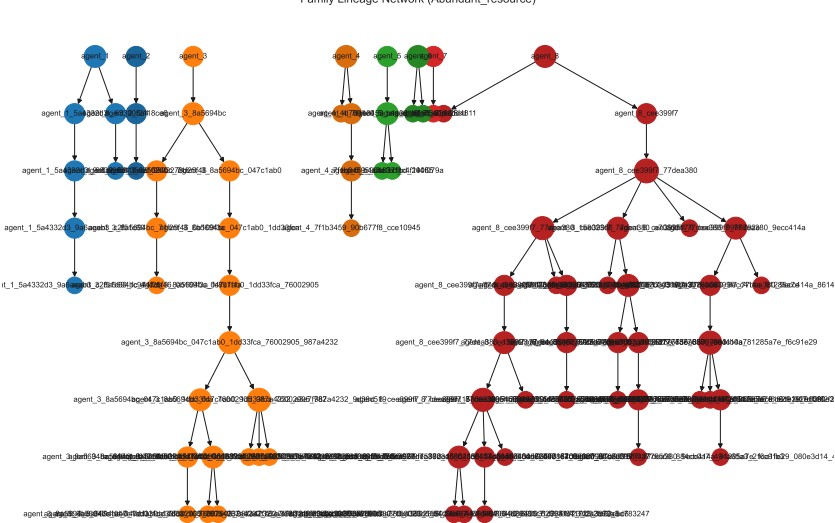

Figure 41: Family Lineage Network for Abundant Resource

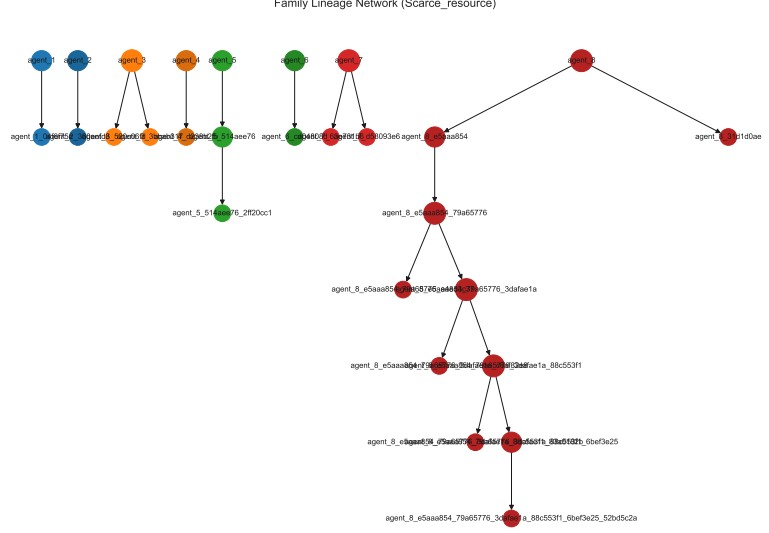

Figure 42: Family Lineage Network for Scarce Resource

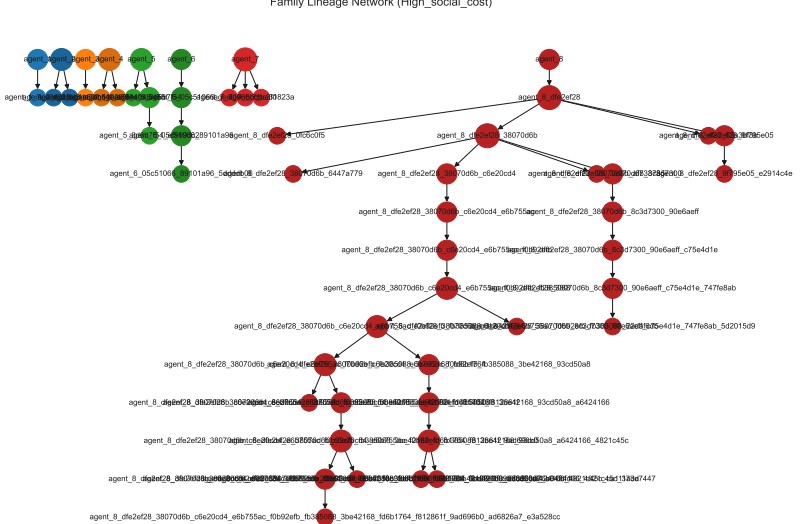

Figure 43: Family Lineage Network for High Social Cost

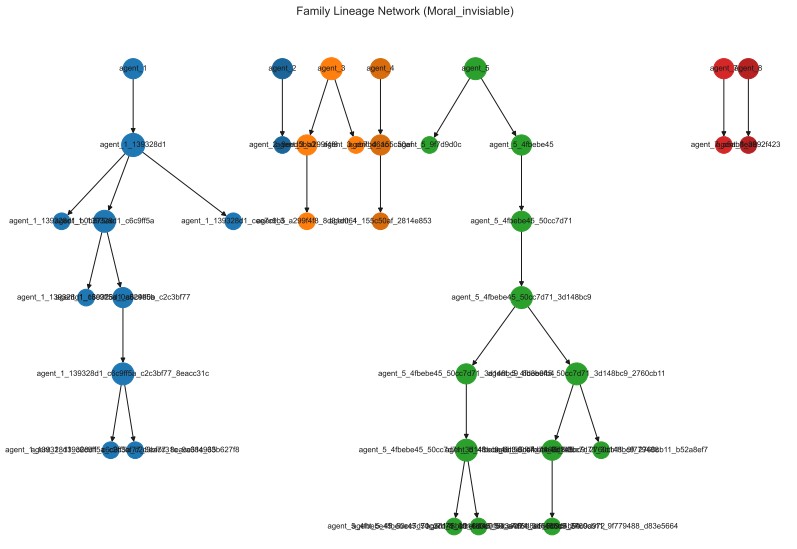

Figure 44: Family Lineage Network for Moral Invisible

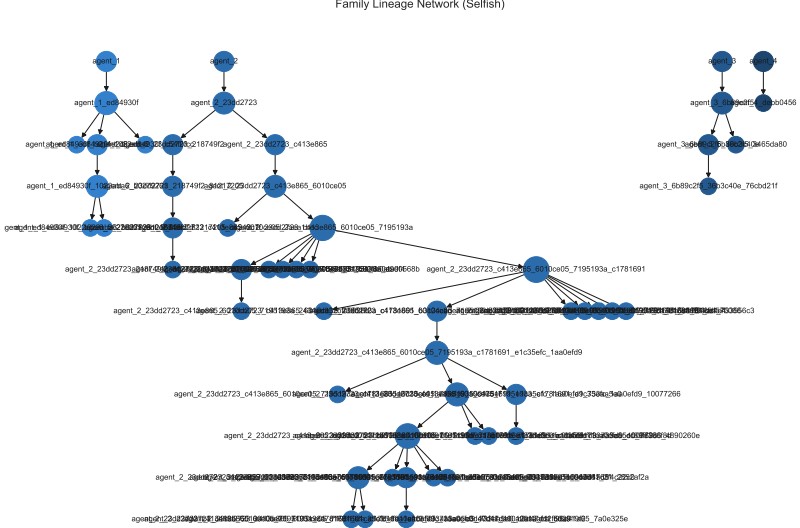

Figure 45: Family Lineage Network for Selfish

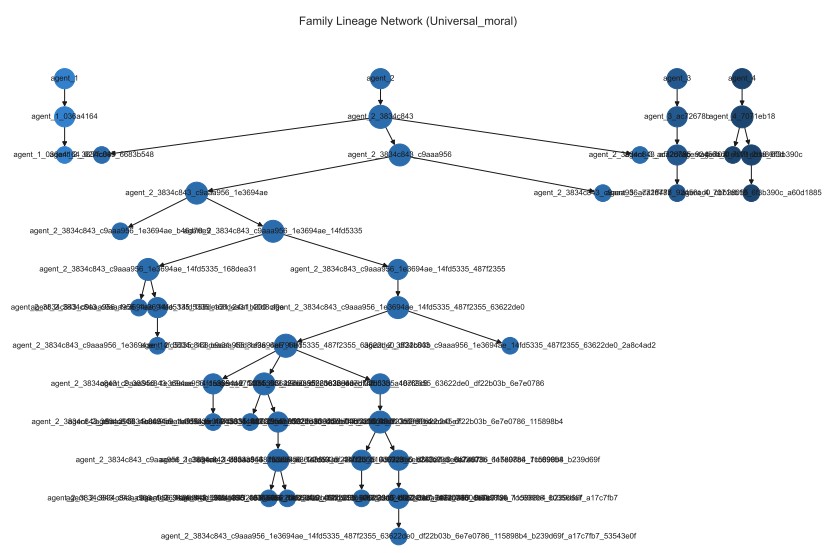

Figure 46: Family Lineage Network for Universal Moral

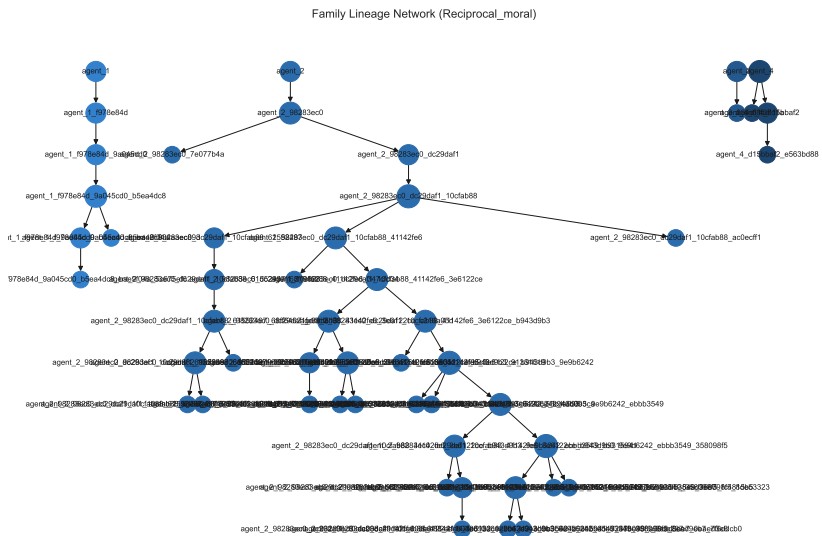

Figure 47: Family Lineage Network for Reciprocal Moral

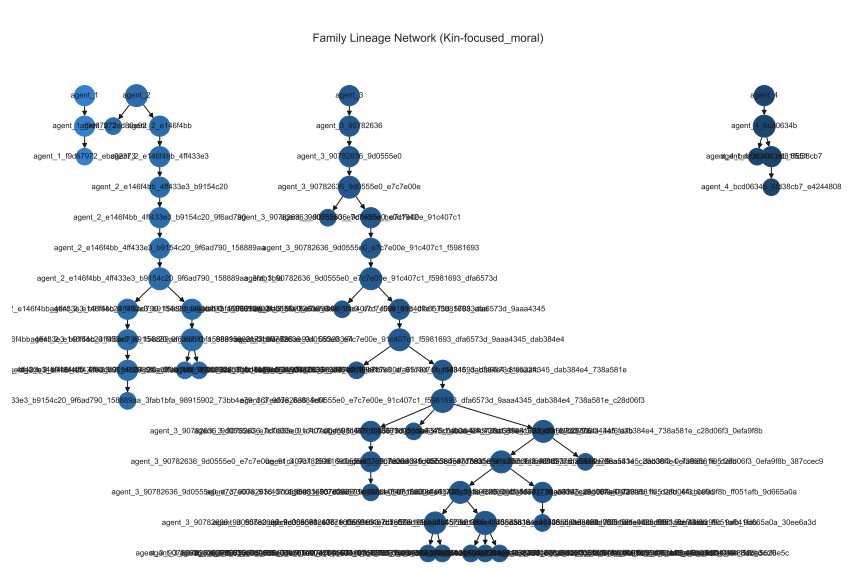

Figure 48: Family Lineage Network for Kin-focused Moral

### G.2.8   Communication network

Figures 49 to 57 depict the communication networks for agents under various scenarios. Each figure uses a network graph where nodes represent agents, and edges represent communication links. Edge thickness may indicate the frequency or strength of communication. Node colors and sizes may represent agent types or influence.

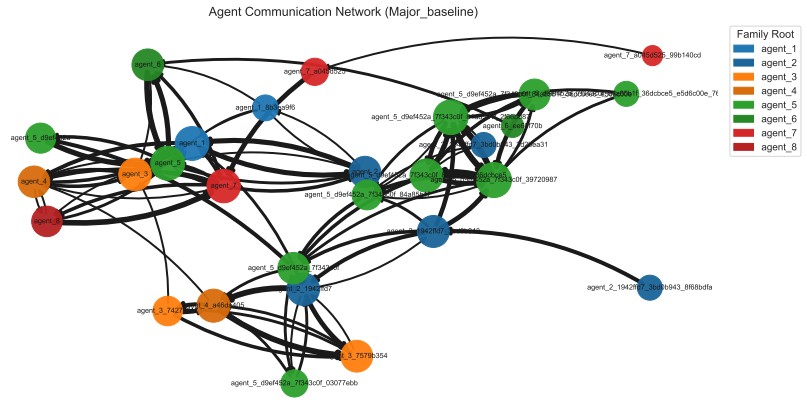

Figure 49: Communication network for Major Baseline

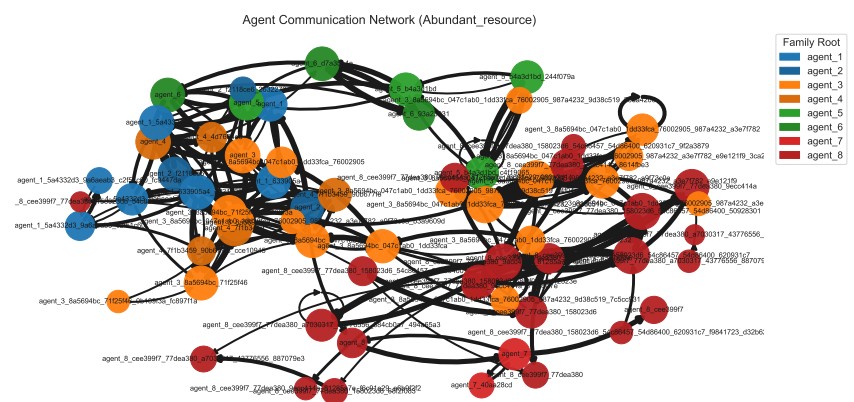

Figure 50: Communication network for Abundant Resource

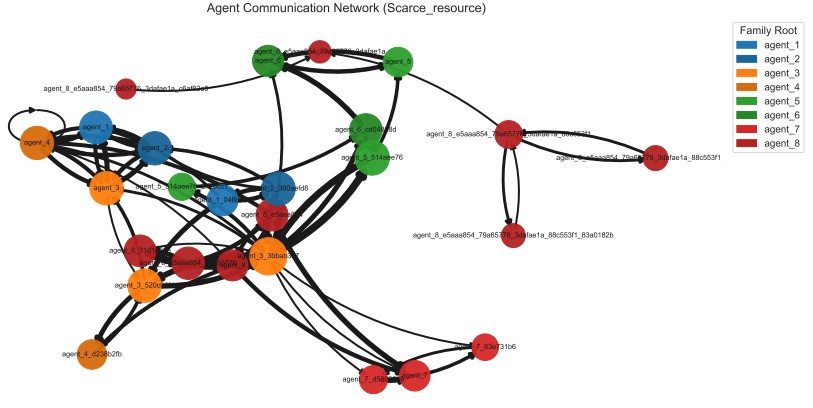

Figure 51: Communication network for Scarce Resource

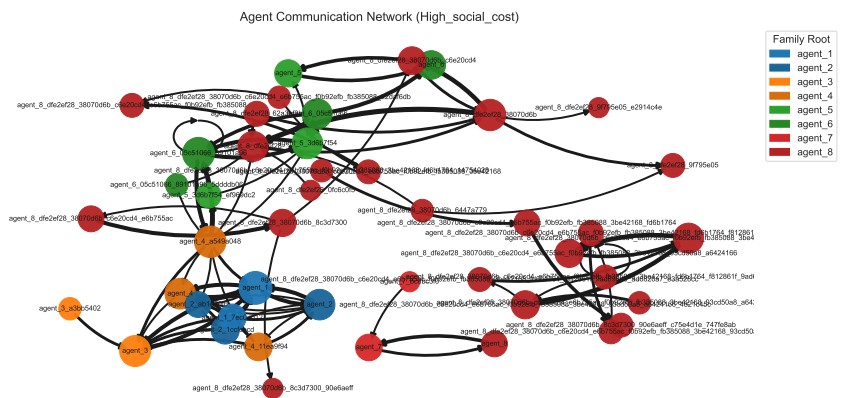

Figure 52: Communication network for High Social Cost

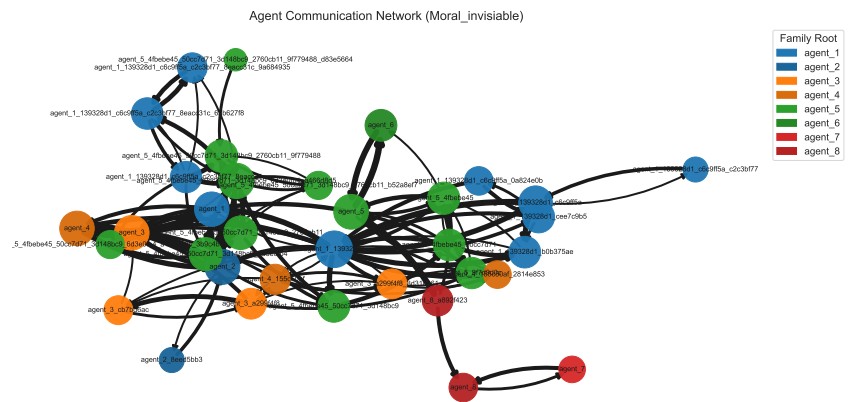

Figure 53: Communication network for Moral Invisible

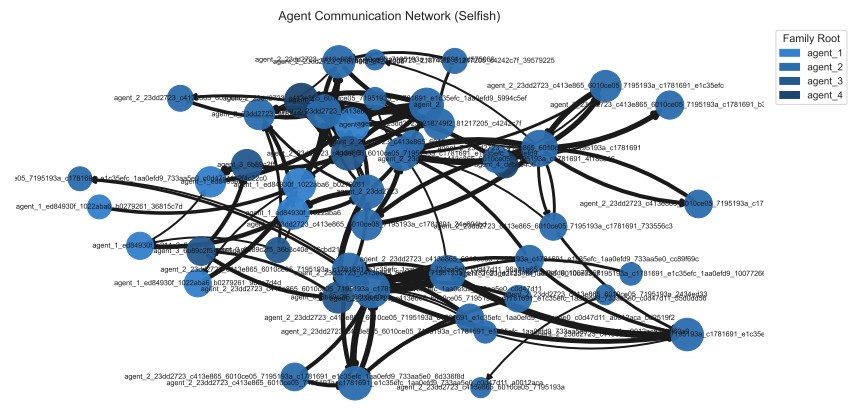

Figure 54: Communication network for Selfish

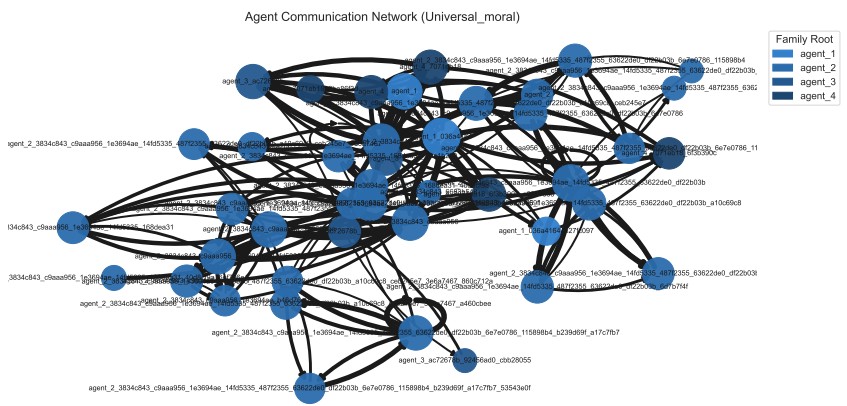

Figure 55: Communication network for Universal Moral

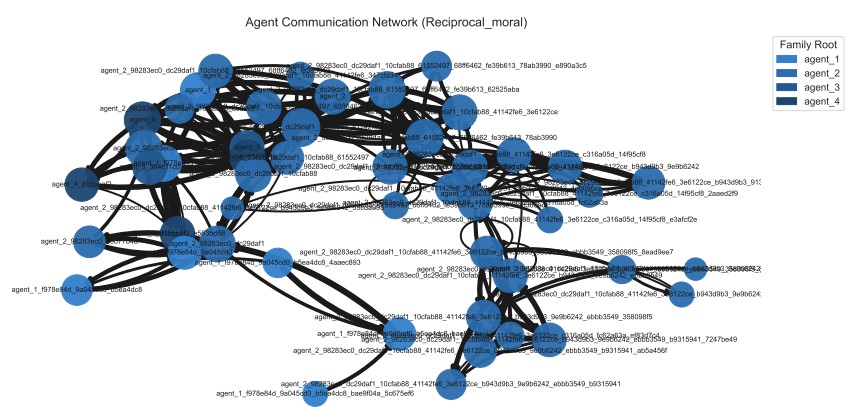

Figure 56: Communication network for Reciprocal Moral

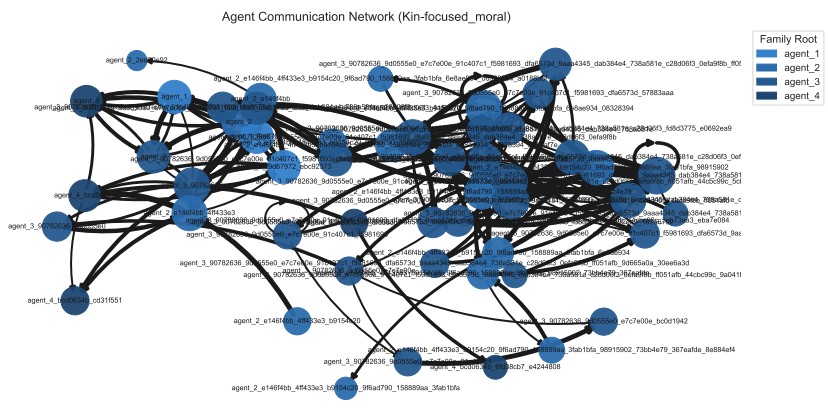

Figure 57: Communication network for Kin-focused Moral

### *G.2.9   Selected hunt collaboration*

Figure  illustrates the selected collaboration dynamics in the major baseline scenario. Each figure shows distributions of the damages and HP allocation of a hun. The x-axis represents

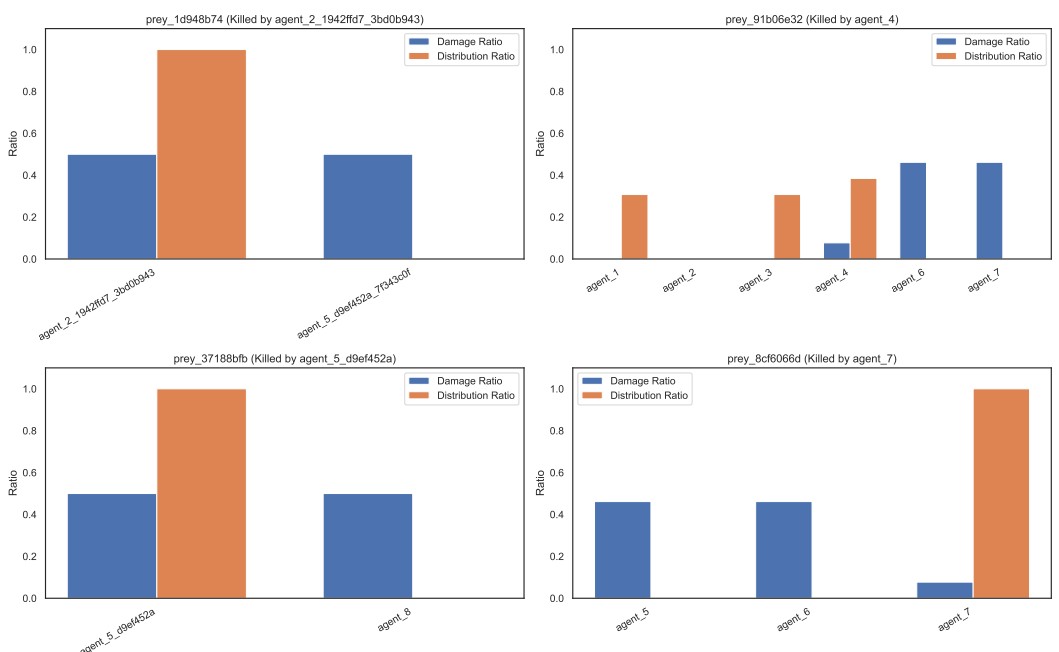

Figure 58: Selected prey hunt and HP distribution for major baseline

the participant agents, while the y-axis represents the ratio of the damages that agents made, and the allocated HP from the killer agent.

