# OpenReview forum: "Investigating Moral Evolution Via LLM-Based Agent Simulation"
_colmweb.org/COLM/2025/Workshop/Social_Sim — Social Sim'25_

### Official Review · Reviewer_CV9t · 2025-07-12
**This paper presents an innovative and methodologically LLM-based simulation to understanding moral evolution. Their findings provide theoretical and empirical insights despite some concerns regarding the design simplicity, lack of clarity in evaluation, and shallow analysis.**

**Rating:** 5
**Overall Assessment:** 3
**Confidence:** 3

**Review:**

**Quality:** The paper demonstrates high methodological rigor. The authors have developed an innovative simulation framework leveraging LLM to simulate complex moral evolutionary dynamics. The validation experiments robustly confirm the alignment between agents' behaviors and their assigned moral dispositions, enhancing confidence in the results.

**Clarity:** The paper is mostly clearly written and structured logically, making the concepts, methodology, and outcomes easily accessible. However, some improvements could be made to improve the clarities in evaluations (e.g., evaluation details, figures, etc)

**Originality:** The paper is highly original, representing a novel integration of LLM-based cognitive modeling with traditional evolutionary frameworks.

**Significance**: The paper's contributions are highly significant and relevant to Social Simulation. The proposed simulation framework holds broad applicability beyond morality alone, potentially providing a valuable new tool for investigating numerous other complex social evolutionary phenomena.

**Comments Suggestions And Typos:**

- Including more diverse morality attributes (e.g., adding individual features) may enhance the overall quality.
- For evaluation and analysis, the authors should include more details about their design and justification to make their results more reliable.

**Paper Summary:**

This paper introduces a novel framework utilizing Large Language Model (LLM)-based agent simulations to explore the evolution of morality within simulated prehistoric hunter-gatherer societies. It specifically examines how agents with varying moral dispositions—selfish, kin-focused, reciprocal moral, and universally moral—interact under different environmental and cognitive conditions. The study offers methodological innovation through the incorporation of psychological realism into evolutionary simulations. It identifies key factors such as resource availability, communication costs, and moral observability that influence the evolutionary outcomes of morality.

**Relevance:**

5

**Summary Of Strengths:**

- The authors use LLM-based simulations to realistically model moral behaviors and interactions, which effectively integrates psychological insights into evolutionary models

- The authors clearly demonstrate that simulated agents behave consistently with their assigned moral types. This alignment is validated through clear statistical evidence.

- Their simulation framework is well-designed and can be potentially extended to other research questions in social evolution beyond morality.

**Summary Of Weaknesses:**

- The setup of morality types is simplistic. There is no clear definition of which group holds stronger influence, nor is it explained how group composition might shape individual intent within the same group or impact interactions with other groups. On the other hand, all individuals in the group are assumed to share a uniform moral type, which lacks diversity and may not accurately reflect the complexities of real-world moral evolution.
- The agents operate in a non-spatial, non-localized environment, which oversimplifies the role of proximity and territory in moral evolution and cooperation.
- The authors attempted to validate the reliability of their simulated moral types through a series of validation experiments. However, the lack of methodological detail hinders the interpretability of the results. For instance, while they state that GPT-4.1 was used for a moral type inference task, it remains unclear what inputs were provided or whether the inference was based on interactions with other moral types or solely based on individual behavior.
- Figures included in the analysis are hard to interpret. Including a good explanation for each figure in the caption or separating complex figures into multiple subfigures for each evaluation aspect may help.
- The analysis relied on LLM as well, but there is a lack of justification for the quality of LLM evaluation results. This makes the results somewhat unreliable.

---

### Official Review · Reviewer_XZEK · 2025-07-17
**LLM-Based Agent Simulation for Modeling Moral Evolution**

**Rating:** 6
**Overall Assessment:** 3
**Confidence:** 3

**Review:**

**Quality:** The paper is technically sound, well-designed, and implements multiple controlled experiments with visualizations, but it lacks comparative baselines with traditional evolutionary models and variance estimates across random seeds.

**Clarity:** The writing is generally clear and well-written, but Figures 2 and 3 are too small and cluttered, making it difficult to analyze without zooming in, and there’s little explanation in the captions to guide interpretation.

**Originality:** To my knowledge, this is the first study to integrate GPT-style cognition with evolutionary moral modeling. This novel approach, leveraging LLMs in a dynamic agent-based setting, stands out as highly original.

**Significance:** The framework is influential for both AI multi-agent research and computational social science. Significance of the work would increase with stricter inferential testing and a comparison against simpler baselines.

**Comments Suggestions And Typos:**

Some key details of the experiment, such as prompt structure and memory configuration, can be moved from the appendix to the main paper to improve clarity on the experiments.

**Paper Summary:**

This paper introduces a novel framework that uses Large Language Models (LLMs) to simulate the evolution of morality in prehistoric hunter-gatherer societies. The agents are assigned one of four distinct moral types based on the expanding circle model. The authors highlight four key contributions:

1. **Methodological contribution:** LLM-based agent simulations for evolutionary moral modeling, incorporating memory and reflection for realistic behavior.

2. **Theoretical contribution:** Insights into how moral dispositions interact with environmental and cognitive factors to produce different evolutionary outcomes.

3. **Empirical contribution:** Evidence on moral type success under different simulated conditions.

4. **Programmatic contribution:** Release of an extensible framework enabling future social evolutionary studies.

**Relevance:**

4

**Summary Of Strengths:**

**Methodological novelty:** This is the first systematic attempt to use LLM-based agents, equipped with memory and reflection, to model and explore moral evolution in a simulated environment.

**Comprehensive experiments:** The study carefully varies key ecological and cognitive parameters such as resource scarcity, communication cost, and moral observability to analyze how different moral strategies perform.

**Empirical validation:** The authors use a soft confusion matrix to show that agent behaviors consistently align with their assigned moral types, supporting the reliability of their simulation setup.

**Programmatic contribution:** The authors present an extensible simulation framework (MoRE) and environment (SOCIAL-EVOL) to support future research on social evolution, including norms, reputation, and group dynamics.

**Summary Of Weaknesses:**

- Each group assumes that all agents share the same moral trait, which removes diversity. In real life, people have mixed values, so the setup doesn’t fully capture how moral differences might evolve or interact.

- The paper only uses one LLM (GPT-4.1-mini). Testing the same setup with different state-of-the-art models could help check if the results hold up more generally.

- The paper doesn’t include a sensitivity analysis to explore how changes in parameters (e.g., resource abundance, social interaction costs, or moral type ratios) affect outcomes. Without this, it’s unclear how robust the findings are or which factors most strongly drive moral evolution.

- The experiments lack important structural and design details like how often agents interact or how long each simulation lasts. Without this, it’s hard for readers to fully understand or evaluate the methods used.

- The paper does not talk about possible issues like hallucinations or strange outputs caused by the prompt. These could affect how the agents behave, so it’s important to mention them.

- The plots are hard to read because they’re cramped and not well-labeled. There’s very little explanation provided, which makes it difficult to understand what the graphs are showing without zooming in.

---

### Official Review · Reviewer_QXZN · 2025-07-18
**Review of Submission 11**

**Rating:** 7
**Overall Assessment:** 4
**Confidence:** 4

**Review:**

See below

**Comments Suggestions And Typos:**

N/A

**Paper Summary:**

The authors present a framework that combining text-based hunter-gatherer environment (social-evol) with LLM-driven agents (more), in which agents are assigned one of four moral dispositions. Each agent then carries an entity-oriented memory, planning and reflection loop, and interacts through hunting, communication, allocation, reproduction and conflict. The authors track population dynamics and behavior to validate that an LLM-judge can recover agentic ground-truth types with high accuracy

**Relevance:**

4

**Summary Of Strengths:**

The pipeline introduced is interesting and can be used by other researchers for similar social-evolution studies. I liked that the framework integrates reflection and memory and goes beyond simple prompting.

**Summary Of Weaknesses:**

The LLM judge is from the same family as the generator models, so high accuracy may only be reflecting the models own beliefs, rather than an actual outcome. Also no comparison to classical evolutionary game simulations or human validation is provided.

---

### Meta-Review · Area_Chair_AcUS · 2025-07-21

**Recommendation:** Accept

**Metareview:**

--